# Coordinated collective migration and asymmetric cell division in confluent human keratinocytes without wounding

Emma Lång[1,2], Anna Połeć[1,2], Anna Lång[1,2], Marijke Valk[3], Pernille Blicher[4], Alexander D. Rowe[1], Kim A. Tønseth[5,6], Catherine J. Jackson[1,5,7], Tor P. Utheim[1,5,7,8], Liesbeth M.C. Janssen[3], Jens Eriksson[1,2] & Stig Ove Bøe[1,2]

Epithelial sheet spreading is a fundamental cellular process that must be coordinated with cell division and differentiation to restore tissue integrity. Here we use consecutive serum deprivation and re-stimulation to reconstruct biphasic collective migration and proliferation in cultured sheets of human keratinocytes. In this system, a burst of long-range coordinated locomotion is rapidly generated throughout the cell sheet in the absence of wound edges. Migrating cohorts reach correlation lengths of several millimeters and display dependencies on epidermal growth factor receptor-mediated signaling, self-propelled polarized migration, and a $G_1/G_0$ cell cycle environment. The migration phase is temporally and spatially aligned with polarized cell divisions characterized by pre-mitotic nuclear migration to the cell front and asymmetric partitioning of nuclear promyelocytic leukemia bodies and lysosomes to opposite daughter cells. This study investigates underlying mechanisms contributing to the stark contrast between cells in a static quiescent state compared to the long-range coordinated collective migration seen in contact with blood serum.

[1] Department of Medical Biochemistry, Oslo University Hospital, 0372 Oslo, Norway. [2] Department of Microbiology, Oslo University Hospital, 0372 Oslo, Norway. [3] Department of Applied Physics, Eindhoven University of Technology, 5600 MB Eindhoven, Netherlands. [4] Department of Medical Biochemistry, Institute of Clinical Medicine, University of Oslo, 0450 Oslo, Norway. [5] Department of Plastic and Reconstructive Surgery, Oslo University Hospital, 0372 Oslo, Norway. [6] Institute of Clinical Medicine, Faculty of Medicine, University of Oslo, 0450 Oslo, Norway. [7] Department of Oral Biology, Faculty of Dentistry, University of Oslo, 0372 Oslo, Norway. [8] Department of Ophthalmology, Oslo University Hospital, 0372 Oslo, Norway. Correspondence and requests for materials should be addressed to S.O.Bøe. (email: stig.ove.boe@rr-research.no)

A migrating epithelial cell sheet is a highly polarized environment where cells coordinate their movements through cadherin-mediated interactions[1,2], cytoskeleton rearrangements, and release of chemokines[3,4]. Integrins link individual cells to the extracellular matrix, providing traction that drives the entire cell sheet[5,6].

Migrating cell sheets involved in wound repair are mainly formed by keratinocytes derived from the basal cell layer of epidermis[7–9], although a recent study also suggests the occurrence of suprabasal cells moving into basal positions[10]. Except for the relatively infrequent cell divisions required for skin homeostasis, basal keratinocytes are mostly dormant under normal physiological conditions. Upon wounding, a number of factors, including calcium, disruption of electrostatic gradient, mechanical tension, and serum exposure, transform keratinocytes from a resting (quiescent) state into a migratory and proliferative state[11–13]. Migration and proliferation is thought to be highly coordinated during the process of re-epithelialization. This is suggested by a recent study in mice showing that a pattern consisting of migrating, non-proliferating cells in the front, a proliferating stationary zone at the back (away from the wound edge), and proliferating migrating cells in the middle arise after wounding[7].

Collective migration of epithelial cells has been extensively studied in several in vivo model systems, such as the *Drosophila melanogaster* border cells, the zebrafish lateral line, the mammalian cornea, and the mouse epidermis[7,14]. Typical in vitro studies involve introduction of a cell-free area in a confluent two-dimensional monolayer, either by scratching (scratch assay) or by removing an obstacle (barrier assay)[15,16]. Subsequently, cell movement is monitored by live microscopy as the cells migrate towards the cell-free area. Collective migration of epithelial cells can also be stimulated in the absence of an artificial wound by the use of electric fields or by unjamming[17–19].

In the present study, we demonstrate that long-range collective migration can be activated in confluent sheets of cultured human keratinocytes through consecutive serum deprivation and serum re-stimulation. These manipulations mimic serum-induced activation of quiescent keratinocytes and lead to long-range coordinated collective migration followed by globally polarized asymmetric cell divisions. Experimental manipulation of the system combined with numerical simulations suggests that persistent long-range coordinated motility is achieved through activation of self-propelled motions guided by a standard Vicsek-like alignment mechanism where each particle in a collective assumes the average direction of motion of the particles in their neighborhood[20]. The study provides insight into the static-to-migratory phase transition that characterizes keratinocytes subjected to wound-induced activation.

## Results

### Activation of collective migration in quiescent cell sheets.

Blood serum contains several essential wound healing factors, including growth factors, cytokines, and anti-microbiotic components[21]. Under normal physiological conditions most epidermal cells reside in a $G_0$ resting state, and wounded tissue is brought into contact with blood at early stages after injury due to bleeding and increased blood vessel permeability. In an attempt to recapitulate these physiological features in an in vitro cell culture system, we used HaCaT keratinocytes. This cell line is derived from human epidermis and has previously been shown to form stratified layers resembling human skin as well as two-dimensional epithelial sheets in culture[22,23]. Cells were grown to 90% confluence and brought into quiescence by 3 days of serum deprivation[24]. Subsequently, the confluent two-dimensional cell sheets were re-stimulated with serum,

and the cell migratory responses were monitored by large-scale live imaging using optical fields (OFs) of up to $8 \times 8$ mm. Particle image velocimetry (PIV) analysis of the data revealed a transient burst of cell sheet motility that extended throughout the entire cell sheet (Fig. 1a; Supplementary Movie 1)[25,26]. Maximal cell velocity, which on average reached 37 µm/h, was observed between 8 and 16 h post stimulation (Fig. 1b). Cell division, as determined by flow cytometry-based quantification of mitotic cells using an antibody against phosphorylated histone H3, was initiated between 25 and 30 h following stimulation (Fig. 1c).

To analyze the time-dependent size dynamics of migrating collectives after serum stimulation, we designed a computer algorithm that calculates the maximal size of contiguous velocity fields (the correlation length, $r$) with velocity vector deviations <90°. Analysis using this software revealed rapid generation of migrating collectives reaching sizes that spanned several millimeters (Fig. 1d). This result suggests that the presence of serum rapidly generates migrating cohorts comprising hundreds of thousands of individual cells migrating collectively in the same direction.

While migration of serum-stimulated quiescent cells was persistent and highly coordinated, unsynchronized confluent cell cultures (before serum starvation) and confluent sheets of starved cells (post starvation but before serum replacement) mostly exhibited modest non-coordinated motilities (Supplementary Fig. 1a; Supplementary Movie 2). To verify the importance of a quiescent cell state prior to serum stimulation, different starvation lengths prior to serum stimulation were tested. This experiment showed that serum starvation for 48 h is required in order to generate the full effect on collective movement, although a slight increase in migration was observed also after 12 and 24 h starvation (Fig. 1e).

We next wanted to identify the primary cell signaling receptor responsible for the serum-activated global migration pattern. Previous studies have shown that migration in several cell types, including HaCaT cells, is activated by the epidermal growth factor (EGF)[27]. In addition, the EGF receptor (EGFR) signaling pathway has previously been implicated in skin homeostasis and wound healing[27,28]. In agreement with this, we found that the EGFR-specific inhibitors lapatinib and gefitinib abolished serum-induced cell sheet motion (Fig. 1f; Supplementary Movie 3). Serum activation and drug-mediated inhibition of EGFR were verified by western blotting using an antibody specific for phosphorylated EGFR (Supplementary Fig. 1b). Consistent with these results, we also found that recombinant EGF alone is sufficient for activation of cell sheet motility following starvation-induced dormancy (Fig. 1g).

### Polarized self-propelled cells drive collective migration.

Since the collective motions in serum-activated dormant cell sheets require EGFR-mediated signaling, the forces that generate motility in this system is likely to stem from EGF-induced polarized propulsion derived from individual cells present within the confluent monolayer. To verify this, we activated migration in quiescent cell sheets in the presence of a serum-free keratinocyte culture medium (CNT-Prime) supplemented with recombinant EGF and varying concentrations of calcium. Previous studies have shown that calcium plays a critical role in promoting epithelial cell–cell adhesion[29–32], and by depleting it, cell motion can be viewed under conditions of strongly reduced cell-to-cell connective forces. For these experiments, average velocity within microscopy fields was calculated based on particle tracking, while migration order was derived from PIV data using a previously described instantaneous order parameter (IOP)[18]. Depletion of

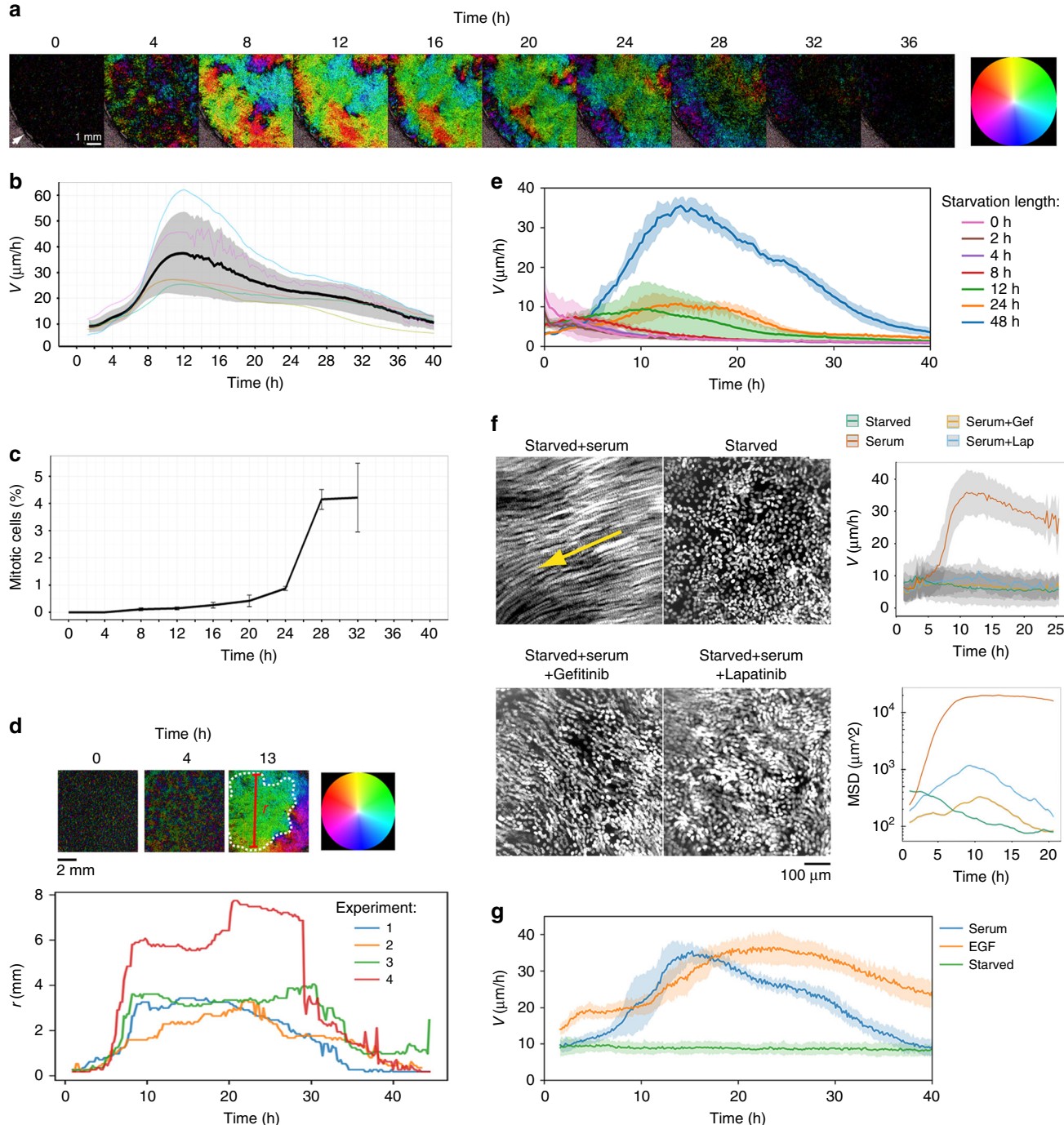

**Fig. 1** Serum-stimulated collective migration in quiescent cell sheets. **a** Serum-induced re-stimulation of HaCaT keratinocytes after starvation. The cell sheet is visualized by particle image velocimetry (PIV). Colors indicate direction of movement, while color intensity indicates migration speed. White arrow in the first time point marks the boundary of the well. The data are extracted from Supplementary Movie 1. **b** Collective migration velocity calculated from the PIV data. The black line represents mean velocity ± SD; $n = 5$ experiments. Optical fields (OFs) = 8 × 8 mm. **c** Activation of cell division after serum-mediated re-stimulation. The percentage of mitotic cells was determined by flow cytometry. The graph represents mean values ± SD; $n = 3$ experiments. **d** Time-dependent correlation length ($r$) of migrating epithelial cell sheets. OF = 8 × 8 mm; $n = 4$ experiments. **e** Serum activation following different serum deprivation lengths. Average velocity is calculated based on the PIV data. Mean ± SD are shown; $n = 3$ experiments. **f** Cell motility in serum-stimulated cells treated with or without EGFR inhibitors. Motion is illustrated by cell trajectories formed by projections of 15 time points comprising the period between 9 and 13 h post serum stimulation. Serum was introduced at time point 0. Yellow arrow indicates direction of migration. Images are extracted from Supplementary Movie 3. Graphs show mean velocity ± SD (upper panel) and mean square displacement (MSD) (lower panel) of cell motions. **g** Quiescent cell sheets were stimulated using EGF (10 ng/ml) or serum. Starved controls were left untreated. Mean velocity was calculated from PIV data. Mean ± SD are shown; $n = 48$ microscopic fields obtained from three separate experiments. OF = 0.65 × 0.65 mm

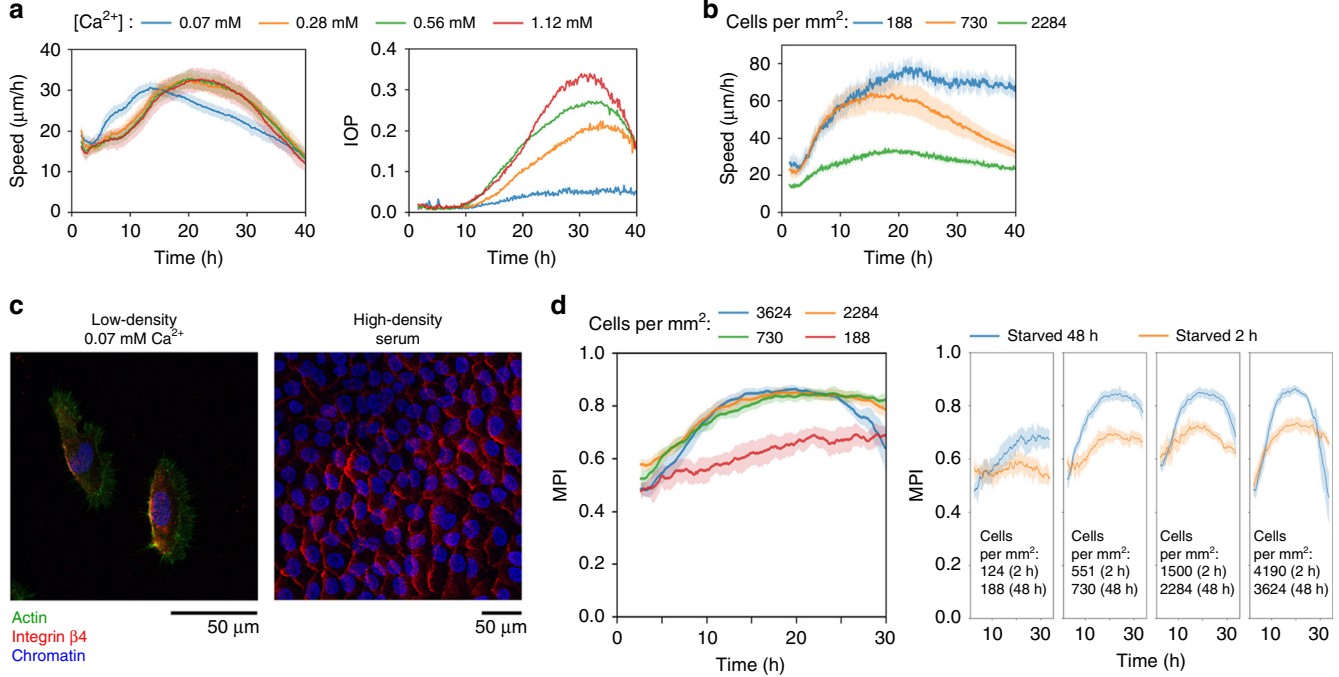

**Fig. 2** Polarized self-propelled cells drive collective migration. **a** Quiescent cell sheets were stimulated in the presence of EGF and varying calcium concentrations. Average speed and IOP were obtained from particle tracking and PIV analysis, respectively. Mean ± SD (speed) and mean values (IOP) are shown; $n = 3$ experiments. **b** Average migration speed generated at different cell densities in the presence of EGF and low calcium concentration. Cells were starved for 48 h prior to stimulation. Mean ± SD are shown; $n = 3$ experiments. Average cell densities at time = 0 are indicated. **c** Integrin β4 polarity was visualized by IF in fixed cells following 48 h of serum deprivation and 16 h of stimulation. Left panel: Cells plated at low density and stimulated in the presence of EGF and 0.07 mM calcium. Right panel: Cells plated at confluent densities and stimulated with serum. **d** Migration persistency index (MPI) over time was generated from particle tracking data. Cells plated at different densities were stimulated in the presence of EGF and 0.07 mM calcium. Left panel: Cells starved for 48 h. Right panels: Comparison of cells starved for 48 h and 2 h. Mean ± SD are shown; $n = 3$ experiments. Average cell densities at time = 0 are indicated

calcium did not significantly affect the maximum migration speed reached (Fig. 2a, left panel; Supplementary Movie 4), but led to a pronounced reduction in ordered migration (Fig. 2a, right panel; Supplementary Movie 4).

Cell-to-cell connectivity was further reduced by plating cells at sub-confluent densities at low calcium concentration (0.07 mM) prior to serum starvation and re-stimulation. Under these conditions, EGF-stimulated cells migrated seemingly independently, but retained a buildup of speed over time that was similar to that observed for collectively migrating cells (Fig. 2b; Supplementary Movie 5).

To visualize the emergence of cell polarity alignment after stimulation, we used integrin β4 and integrin α6, two proteins that play important roles in keratinocyte adhesion and motility, as migration directionality markers[33]. Integrin β4 localized to trailing edges of individually migrating cells, and this sub-cellular localization translated into highly ordered wave-like patterns across large regions of cell sheets during collective migration (Fig. 2c). Notably, these global integrin patterns, which emerged due to directional alignment of migrating cells, were not detected in non-stimulated-starved cells or in cells that had not been subjected to serum deprivation (Supplementary Fig. 2a). Migration directionality relative to the polarized integrin distribution was confirmed by live imaging of cells labeled with fluorescently tagged integrin α6-specific antibodies (Supplementary Movies 6 and 7). Combined, these results show that serum-induced motility of quiescent monolayers is generated through activation and alignment of self-propulsive polarized forces derived from individual cells within the collective.

To investigate the ability of cells to coordinate migration relative to neighboring cells, we made use of a computer algorithm that calculates a migration persistency index (MPI) of single cells over time based on particle tracking data. A low MPI value approaching 0 suggests random directionality between tracked time points, while an MPI value of 1 implies persistent migration in a straight line. Analysis of starved cells following EGF-mediated stimulation at various cell densities under low calcium conditions revealed increased migration persistency at higher densities, suggesting that migrating cells are influenced by neighboring velocity fields (Fig. 2d, left panel; Supplementary Movie 5). As a control, we used non-starved HaCaT keratinocytes which, similar to starved cells, exhibited extensive self-propelled migration under conditions of low calcium concentration and sub-confluent cell density (Supplementary Fig. 2b; Supplementary Movie 8). Notably, we observed significantly higher MPI values in starved compared to non-starved cells following stimulation (Fig. 2d, right panels; Supplementary Movies 5 and 8). These experiments suggest that cells that have been awakened from starvation-induced dormancy by EGF stimulation have a higher capacity for neighbor-induced migration alignment compared to non-starved cells.

**Numeric simulations of serum-induced collective migration.** Based on the experimental data, we produced a numerical simulation model that explicitly takes into account the EGF-induced motility of individual cells, the calcium-dependent connectivity between cells and the confluence of the cell layer. Briefly, we model the confluent cell layer using a Voronoi tessellation and assume that each cell has a preferred cell surface

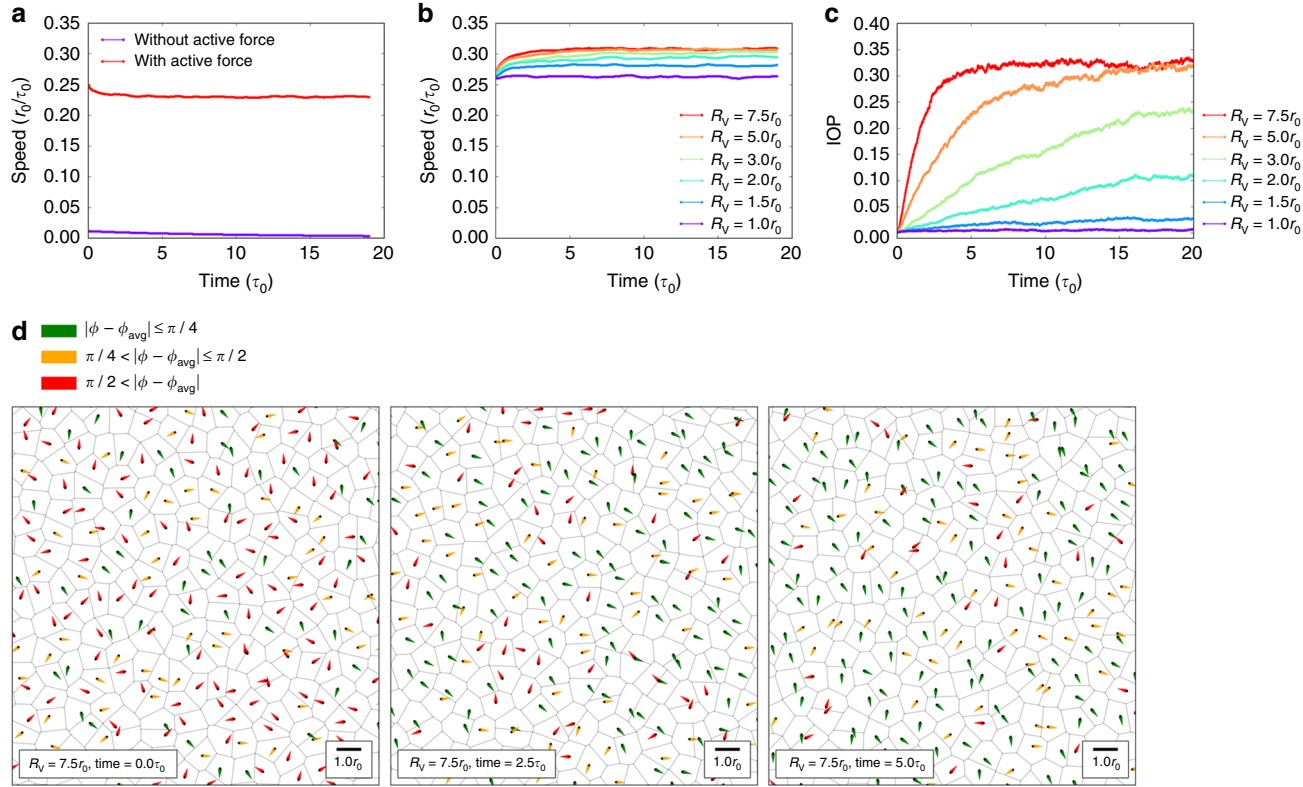

**Fig. 3** Numeric simulations of serum-induced collective migration. **a** Time-dependent average cell speed obtained from simulations in the presence and absence of self-propelling forces. **b** Time-dependent average speed and **c** instantaneous order parameter (IOP) obtained from simulations of active cells with varying Vicsek radii, mimicking the effect of varying cell–cell connectivities. **d** Simulation snapshots at different times $t$ for strongly connected cells with Vicsek radius $R_V = 7.5\ r_0$. The arrows indicate the cells' instantaneous velocity directions, and are color-coded by their deviation from the average velocity direction

area and cell perimeter[34]. Upon EGF stimulation, all cells become active and experience an additional self-propulsion force, which we model with the alignment mechanism of the Vicsek model[20]. In this model, each cell tends to align its self-propulsion direction with the velocity of all its neighboring cells that fall within a distance $R_V$ (the so-called Vicsek radius), in the presence of Brownian noise. The full cell dynamics is thus governed by a balance between the Vicsek alignment mechanism and the tendency to achieve a certain cell perimeter. We incorporate the effect of varying calcium concentrations by changing the Vicsek radius: a Vicsek radius of zero describes fully disconnected cells, while a large Vicsek radius implies strong cell–cell connectivity. Our simulations reveal that the absence of active self-propulsion forces leads to a negligible average cell speed, similar to what is observed in the experiments with starved cells, while the presence of EGF-induced cell motility leads to an average speed of approximately $0.25 r_0/\tau_0$ (Fig. 3a). We also found that an increasing Vicsek radius does not significantly change the speed of the cells (Fig. 3b), with average values ranging between 0.26 and $0.32 r_0/\tau_0$, but it does lead to stronger alignment and flocking behavior (Fig. 3c, d; Supplementary Fig. 3; Supplementary Movies 9–11). This supports the experimental findings that enhanced calcium concentrations lead to a larger IOP index, and implies that the observed collective motion upon EGF stimulation arises from a combination of autonomous single-cell movement and strong inter-cell connectivities. It should be noted that in our simulations we have assumed the motility and connectivity to increase instantaneously at time $t = 0$, causing an immediate increase in alignment, while in our cell culture-based

experimental model the alignment builds up more slowly, likely due to the fact that cells require a finite time to incorporate the physiological effects of EGF and calcium.

**Pre-mitotic nuclear motility and cell division polarity.** Since migration and cell division occurs in temporally overlapping phases, we investigated the relationship between cell migration and cell division polarity. We first examined the spatiotemporal dynamics of mCherry-tagged Histone H2B during mitosis. Analysis of cell division axes revealed an overrepresentation of cell divisions that were oriented parallel to the direction of cell migration (Fig. 4a). We also identified a polarized chromatin pattern at early stages of mitosis in more than 80% of cell divisions analyzed. This pattern was characterized by an early prophase invagination on one side (designated P2) of the nucleus (Fig. 4b). The prophase-specific U-shaped chromatin appears to form in close conjunction with the cell boundaries as the convex side (designated P1) of the structure was observed to be closely aligned with the plasma membrane (visualized by phalloidin) of early prophase cells (Fig. 4c). Furthermore, time-lapse analysis of HaCaT cells stably expressing mCherry-α-tubulin and GFP-Histone H2B, and immunofluoresence (IF) analysis of fixed HaCaT cells using α-tubulin-specific anti-bodies, showed that the nuclear invagination is the site of prophase spindle pole activation (Fig. 4d; Supplementary Fig. 4a). These observations suggest the presence of a cell polarity axis at early stages of prophase.

The biased positioning of chromatin towards the P1 side of the cell early in mitosis suggest that the nucleus might migrate before

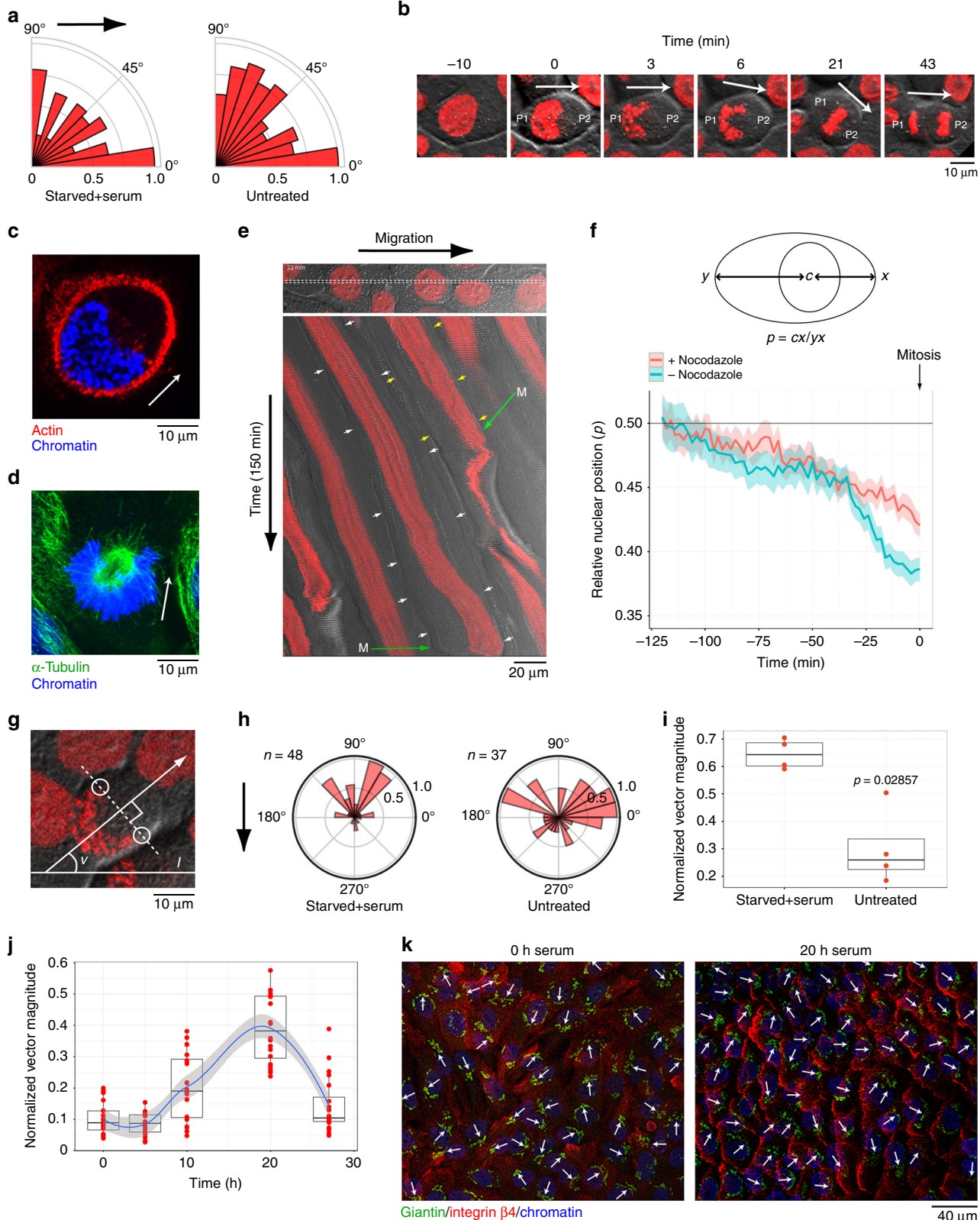

cell division. To investigate this, we tracked mCherry-Histone H2B-labeled nuclei relative to the plasma membrane for 2 h prior to mitosis (Fig. 4e; Supplementary Movie 12). As a control we used HaCaT cells cultured in the presence of low concentrations of the microtubule inhibitor nocodazole, which reduced plasma membrane proximal positioning of chromatin during prophase.

The nucleus was generally observed to be stably positioned at the cell center in most interphase cells. However, approximately 30 minutes (min) before mitosis entry we observed active movement of the nucleus towards one side of the cell (Fig. 4e, f; Supplementary Movie 12). Furthermore, pre-mitotic nuclear migration was observed to consistently occur in the same

**Fig. 4** Pre-mitotic nuclear motility and cell division polarity. **a** Quantification of cell division angles. Angles were defined relative to the direction of migration (left panel, black arrow, $n = 119$) or relative to an arbitrary line (right panel, $n = 120$). **b** Chromatin configurations predicts the P1 and P2 side of a dividing cell. Snapshots of a living HaCaT cell expressing mCherry-Histone H2B (red) in combination with DIC (gray) are shown. **c** Phalloidin-labeled actin (red) and DAPI (blue) illustrates early prophase chromatin configuration in a PFA-fixed cell. **d** Spindle poles localize to the concave side of prophase chromatin. IF-labeled α-tubulin (green) and DAPI (blue) in a PFA-fixed cell is shown. **b–d** White arrow indicates cell division orientation as defined by the early prophase chromatin configuration. **e** Kymograph of collectively migrating cells entering mitosis. Nuclei and cell boundaries are visualized by mCherry-Histone H2B (red) and DIC (gray), respectively. White and yellow arrows indicate cell boundaries. Green arrows indicate mitosis entry. Images are extracted from Supplementary Movie 12. **f** Quantitative assessment of pre-mitotic nuclear migration. Mean relative nuclear position ($p$) ± SEM over a period of 2 h prior to mitosis entry is shown. Data were collected from two experiments; nocodazole-treated, $n = 50$ cells; untreated, $n = 54$ cells. **g** The prophase angle ($v$) is defined by the division vector (arrow) and a constant arbitrary line ($l$). **h** Radial diagrams showing the distribution of prophase angles ($n = 4$) in serum-stimulated and untreated cells. Data were subjected to unity-based normalization (0–1) before plotting. Black arrow indicates direction of migration. **i** Normalized cell division vector magnitudes in serum-stimulated and untreated cells. The $y$-axis shows the proportion of cell divisions occurring along the dominant orientation; *$p = 0.029$; $n = 4$ experiments. **j** Serum-induced nucleus-to-Golgi alignment. Normalized nucleus-to-Golgi polarity vector magnitudes at indicated time points after serum stimulation. Superimposed is a regression line (blue), with 95% confidence interval; $n = 2$ experiments. **i–j** Black lines are the median, whiskers indicate min and max values. **k** Nucleus-to-Golgi alignment (white arrows) superimposed on IF-labeled integrin β4 (red) illustrates alignment with migration direction after serum stimulation

direction as the collective cell migration (Fig. 4e; Supplementary Movie 12).

We next compared cell division polarity, which we defined based on the prophase chromatin configuration as depicted in Fig. 4g, in untreated cells and cells subjected to serum deprivation and re-stimulation. Serum stimulation resulted in a majority of cell divisions being oriented opposite to the direction of collective cell migration, while mitotic orientations in untreated asynchronous HaCaT keratinocytes were mostly random (Fig. 4h; Supplementary Figs. 4b and 5; Supplementary Movies 13 and 14). Statistical analysis using a non-parametric Wilcoxon's rank-sum test revealed a strong correlation between serum treatment and globally oriented mitosis polarity (Fig. 4i, *$p = 0.029$).

To reconcile serum-induced global migration polarity with global cell division polarity, we investigated relative positioning of the Golgi apparatus and the nucleus during collective migration[35]. We observed a gradual buildup of unidirectional alignment between 10 and 20 h following serum stimulation and a significant drop in alignment after 27 h (Fig. 4j, k; Supplementary Fig. 4c). Comparison to integrin β4 staining 20 h after stimulation showed that cells primarily adopt a nucleus-to-Golgi configuration where the Golgi faces the migrating cell front (Fig. 4k). These results suggest that the nucleus reverses its position relative to microtubule-organizing center (MTOC) prior to cell division. We confirmed this by examination of 15 early prophase cells immunofluorescently stained with antibodies specific for integrin β4 and giantin. Reversed nucleus-to-Golgi polarity relative to migration was observed in all examined cases (Supplementary Fig. 4d).

**Asymmetric inheritance of PML bodies.** PML bodies are nuclear compartments involved in a range of cellular processes, including senescence, differentiation, growth regulation, and genome maintenance[36]. The main constituent of PML bodies is the PML protein, which plays a major role in forming a three-dimensional network of protein interactions that constitute the PML body core structure[37,38]. During cell division, PML bodies are inherited by daughter cells through a process that involves release of PML bodies into the mitotic cytoplasm after nuclear membrane breakdown. When cells exit mitosis, PML bodies remain in the cytoplasm for a certain period, where they gradually disassemble into smaller components that become imported by the daughter nuclei for reuse in production of progeny PML bodies. Thus, the cytoplasmic PML bodies that are detected in newly divided $G_1$ cells are structures derived from the pre-mitotic mother nucleus (Fig. 5a)[39,40]. We refer to these $G_1$-specific cytoplasmic bodies as cytoplasmic assemblies of PML and nucleoporins (CyPNs)[40].

Since PML bodies represent nuclear compartments that persist during mitosis, they could potentially become asymmetrically distributed during polarized division of HaCaT keratinocytes. To investigate this, newly divided daughter cell pairs were identified using Aurora B as midbody marker and scored as asymmetric if only one of the daughter cells contained detectable CyPNs (Fig. 5b). The ratio of asymmetric or dual PML body inheritance was determined by counting the number of CyPNs in each daughter cell. The majority (73.8 ± 2.7%) of daughter cell pairs analyzed contained detectable CyPNs in both cells, indicating dual inheritance of PML bodies (Fig. 5b; Supplementary Fig. 6a). A smaller fraction (26.2 ± 2.7%) contained detectable CyPNs in only one cell, suggesting asymmetric partitioning during mitosis (Fig. 5b; Supplementary Fig. 6a, red asterisks). Interestingly, analysis of cells following serum deprivation and 30 h serum stimulation (a time point where the migration and division phases overlap) revealed a significant increase in complete asymmetric PML body inheritance events (45.0 ± 3.7%), compared to asynchronously growing cells (26.2 ± 2.7%) (Fig. 5c, **$p = 0.0021$; Supplementary Fig. 6a, b, red asterisks). The experimental approach used for identification of daughter cell pairs was validated using α-tubulin staining (Supplementary Fig. 6c, d). We also studied asymmetric PML body inheritance in human primary epidermal keratinocytes (HEKn). These cells were observed to have fewer CyPNs compared to HaCaT cells (making statistical analysis of PML body inheritance more difficult), but the ratio of asymmetric inheritance events was similar in the two cell lines (Fig. 5c; Supplementary Fig. 7a, red asterisks). Furthermore, analysis of the tumor cell lines NB4 and HeLa revealed a low ratio of asymmetry, consistent with random PML body distribution (Fig. 5c; Supplementary Fig. 7b, c, red asterisks). Statistical analysis revealed a significantly higher proportion of HaCaT daughter cells with complete asymmetric inheritance of PML bodies than could be expected based on a theoretical binomial distribution of PML bodies (Fig. 5d).

We next examined the spatiotemporal distribution of PML bodies relative to the polarized chromatin configuration observed in dividing HaCaT cells. For this experiment, we performed confocal time-lapse imaging using live HaCaT cells stably expressing EYFP-PML1 and mCherry-Histone H2B (Fig. 5e; Supplementary Movie 15). We identified and tracked 14 cell divisions that exhibited complete asymmetric distribution of PML bodies. The PML bodies were inherited by the P1 daughter cell in all cases examined (Fig. 5f, ****$p = 6.1 \times 10^{-5}$). Analysis of EYFP-PML1 and mCherry-Histone H2B in combination with differential interference contrast (DIC) revealed that the PML bodies

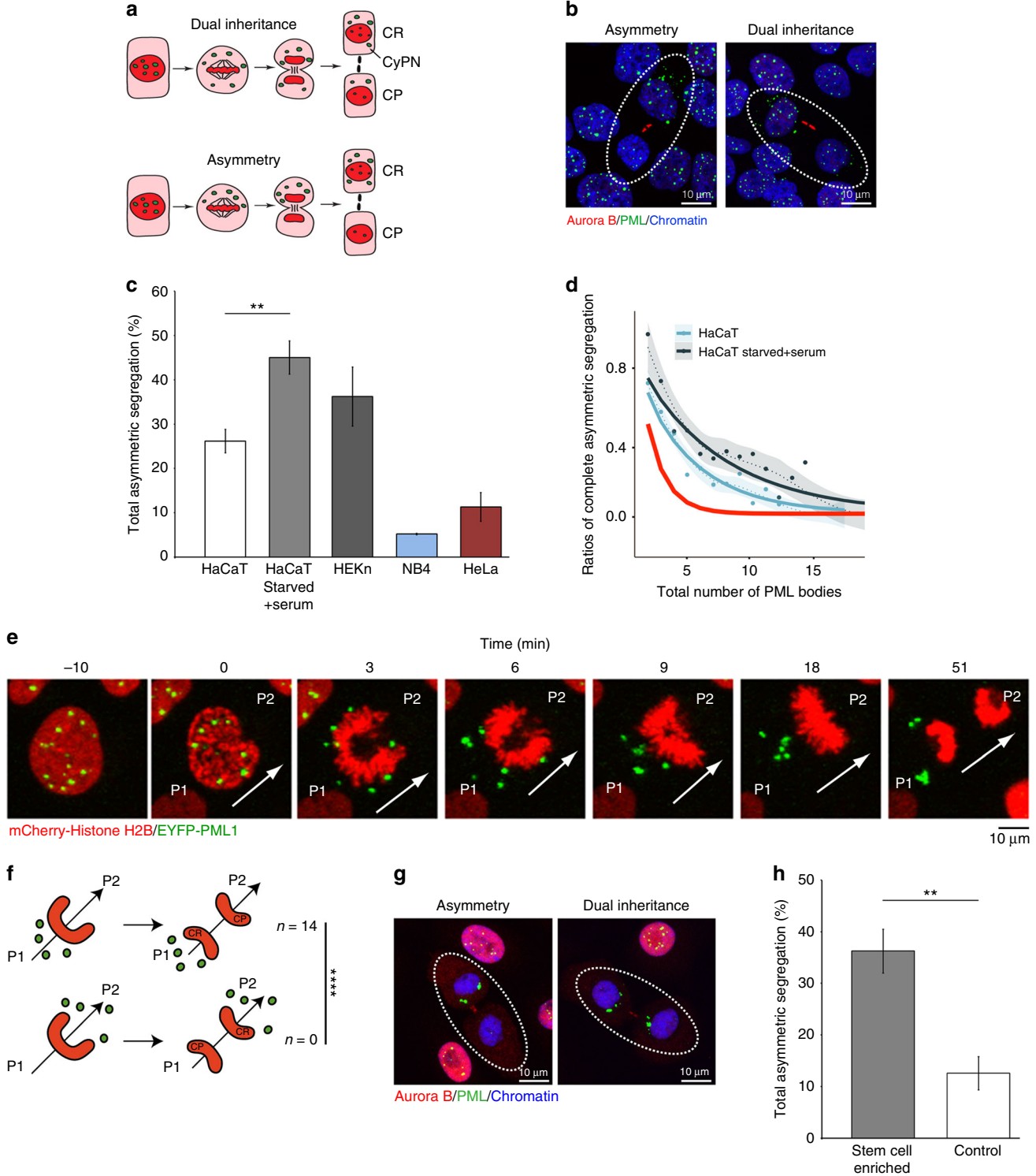

preferentially are released towards the P1 side of the mitotic chromatin following envelope breakdown. Chromatin motility is then reversed towards the center of the cell, leading to asymmetric deposition of PML to the P1 side of the mitotic cell (Supplementary Fig. 8).

To investigate if asymmetric inheritance of PML bodies is related to stemness, we analyzed the PML body distribution in freshly isolated human epidermal keratinocytes (Fig. 5g). Stem cell enrichment, through collagen IV attachment[41], increased the ratio of asymmetric PML body inheritance events significantly

compared to non-fractionated controls (Fig. 5h, **$p = 0.0015$; Supplementary Fig. 9a–c, red asterisks).

**Asymmetric inheritance of lysosomes**. A previous study, using a human mammary epithelial cell line, showed that aged mitochondria become asymmetrically apportioned to daughter cells during cell division. To distinguish between aged and young mitochondria, the authors of this publication employed pulse labeling of Snap-tagged mitochondrial proteins, mostly using

**Fig. 5** Asymmetric **i**nheritance of PML bodies. **a** Schematic of PML body behavior during mitosis. In interphase, PML bodies are mainly localized within the nucleus. During mitosis, PML bodies enter the cytoplasm of newly formed daughter cells and form cytoplasmic assemblies of PML and nucleoporins (CyPNs). We refer to the daughter cell with the highest CyPN number as CyPN rich (CR), while the other is designated CyPN poor (CP). **b** Examples of asymmetric (left panel) and dual (right panel) inheritance of PML bodies in HaCaT cells. Note that all CyPNs (cytoplasmic bodies) localize to only one daughter during asymmetric inheritance. **c** Quantification of total asymmetric segregation events. For each cell line, the average of three experiments ± SD is shown; $**p = 0.0021$. **d** Incidence of cell divisions showing complete asymmetric inheritance in HaCaT cells. Best-fit curves for asynchronously growing (solid blue line; $p_{sym} = 0.20$; $n = 469$) and starved and serum-stimulated (solid black line; $p_{sym} = 0.14$; $n = 298$) cells are shown. Smoothed trend lines are shown as dashed lines with a 95% CI. The theoretically predicted curve for a completely symmetric distribution (solid red line; $p_{sym} = 0.5$) is shown for comparison. **e** Live cell imaging of HaCaT cells expressing EYFP-PML1 and mCherry-Histone H2B. Arrow denotes cell division orientation. Images are extracted from Supplementary Movie 15. **f** Cell divisions exhibiting complete segregation of PML bodies were identified by live cell imaging and tracked. In all cases analyzed, the CR cell corresponded to P1, while the CP cell corresponded to P2; $****p = 6.1 \times 10^{-5}$; $n = 14$. **g** Newly divided daughter cell pairs of human primary epidermal keratinocytes exhibiting asymmetric and dual PML body inheritance, respectively. **h** Quantification of total asymmetric PML body inheritance in human primary epidermal keratinocytes, before and after stem cell enrichment. The average of three experiments ± SD is shown; $**p = 0.0015$

Snap-tagged OMP25 labeled with the TMR-Star fluorophore as a marker of aged mitochondria[42]. We wanted to repeat these experiments in our system to investigate if asymmetric PML body inheritance correlates with stemness and asymmetric segregation of aged mitochondria. This experiment was also motivated by the finding that aged Snap-tagged mitochondrial proteins were intimately associated with the nucleus, possibly implicating nuclear migration and skewed nuclear positioning during nuclear envelope breakdown, in the mechanism of asymmetric mitochondria apportioning[42]. We generated HaCaT cells stably expressing Snap-tagged OMP25 and performed pulse labeling on serum-stimulated quiescent cells using TMR-Star and 647-SiR at 30 and 3 h prior to imaging. We observed asymmetric distribution and perinuclear localization of TMR-Star in mitotic and interphase cells, respectively (Supplementary Fig. 10). However, the fluorophore distribution did not depend on aging, since the same phenomenon was observed when the labels representing aged and young mitochondria were swapped (Supplementary Fig. 10a, b). In addition, TMR-Star, but not 647-SiR, reacted with cytoplasmic vesicles in cells without Snap-OMP25 expression (Supplementary Fig. 10c). Upon further investigation, we found that these TMR-Star-positive vesicles co-localize with the acidotrophic die Lyso-Tracker Green (LTG), suggesting that they were lysosomes (Supplementary Fig. 10d; Supplementary Movie 16). Thus, lysosomes, but not aged mitochondria, become asymmetrically apportioned during division of HaCaT keratinocytes. TMR-Star-labeled vesicles become increasingly concentrated on the centrosome side of the nucleus immediately prior to mitosis and the labeled vesicles preferentially localized to P2-derived daughter cells after cell division (Fig. 6a, b, $****p = 8.8 \times 10^{-5}$; Supplementary Movie 17). A similar pattern was observed for LTG-labeled lysosomes in HaCaT cells devoid of transgene expression (Fig. 6c, d, $****p = 1.8 \times 10^{-6}$; Supplementary Movie 18). These results suggested that lysosomes and PML bodies become distributed to opposite daughter cells during cell division. To confirm this, we analyzed the relative distribution of endogenous lysosomes and PML bodies in newly divided daughter cells by IF in fixed cells. For this experiment we only included cell pairs that could be identified as freshly divided daughters (small paired nuclei and PML bodies exclusively in the cytoplasm). We detected PML and lysosome enrichment in opposite cells in approximately 70% of daughter cell pairs analyzed (Fig. 6e, f, $**p = 0.0026$). Combined, our data demonstrate a polarized mitosis, which is aligned with cell migration and leads to preferential segregation of PML bodies and lysosomes to the front and rear daughter cell, respectively (Fig. 6g).

**Lysosome segregation regulates stemness and cell motility**. To investigate potential differences between daughter cells that arise from asymmetric cell division, we stimulated quiescent

HaCaT keratinocytes with serum for 40 h to obtain a population of cells where more than 90% of the cells had passed through mitosis once. These cells were subjected to LTG labeling and subsequently fractionated into LTG (High) and LTG (Low) cell populations by fluorescence-activated cell sorting (FACS) (Fig. 7a). Western blot analysis of proteins isolated after 2 days in culture revealed high LAMP-1 and LAMP-2 expression in LTG (High) cells showing that these cells maintain a high lysosome content (Supplementary Fig. 11a, b). Conversely, we did not observe differences in PML expression (Supplementary Fig. 11a). We first analyzed the two cell populations by plating them at clonal densities. For single cells, the time elapsed between the first and second cell division was recorded, in order to estimate cell cycle length. In the same experiments, we also determined the extent of two-cell colony rotation, which is a parameter previously used for prediction of human keratinocyte stemness (Fig. 7b)[33]. The analysis revealed a similar mean cell cycle length in LTG (High) (20.8 ± 3.5 h) and LTG (Low) (20.5 ± 3.8 h) cells (Fig. 7c, $p = 0.24$; Supplementary Movie 19). Also, we did not observe significant differences in cell colony size 2 and 6 days after plating (Supplementary Fig. 11c, d). Analysis of two-cell colony rotations, on the other hand, revealed a significantly higher rotation rate during the two-cell colony stage in LTG (Low) (3.0 ± 3.1 rotations) compared to LTG (High) (1.8 ± 2.5 rotations) cells (Fig. 7d, $****p = 9.99 \times 10^{-8}$; Supplementary Movie 19). We next analyzed the presence or absence of stem cell and differentiation markers in cell colonies 6 days after plating. We did not detect significant differences in the number of colonies expressing the differentiation markers cytokeratin 10 (K10), involucrin or loricrin, suggesting that LTG (High) and LTG (Low) cells possess a similar capacity for giving rise to differentiated cells. However, a significantly higher number of LTG (Low) compared to LTG (High) cells gave rise to colonies positive for the stem cell marker cytokeratin 15 (K15), further suggesting that the LTG (Low) population is enriched in cells containing a stem-like phenotype (Fig. 7e, $*p = 0.024$; Supplementary Fig. 11e)[33].

Numerous studies of mitosis in tissue development, tissue homeostasis, and tissue repair have shown that asymmetric cell division gives rise to cells with distinctive migration capacities[43–46]. To analyze the movement of sorted cell populations, we plated cells on collagen IV-coated glass bottom dishes at densities that yield colony sizes of approximately 5 to 60 cells per colony. Live imaging of plated cells revealed higher motility of LTG (High) compared to LTG (Low) cells. Colonies produced by LTG (High) cells, but not those produced by LTG (Low) cells, had the ability to move persistently and directionally on the collagen IV-coated surface (Fig. 7f; Supplementary Movie 20). To quantify cell motility we recorded the trajectories of randomly selected cells for

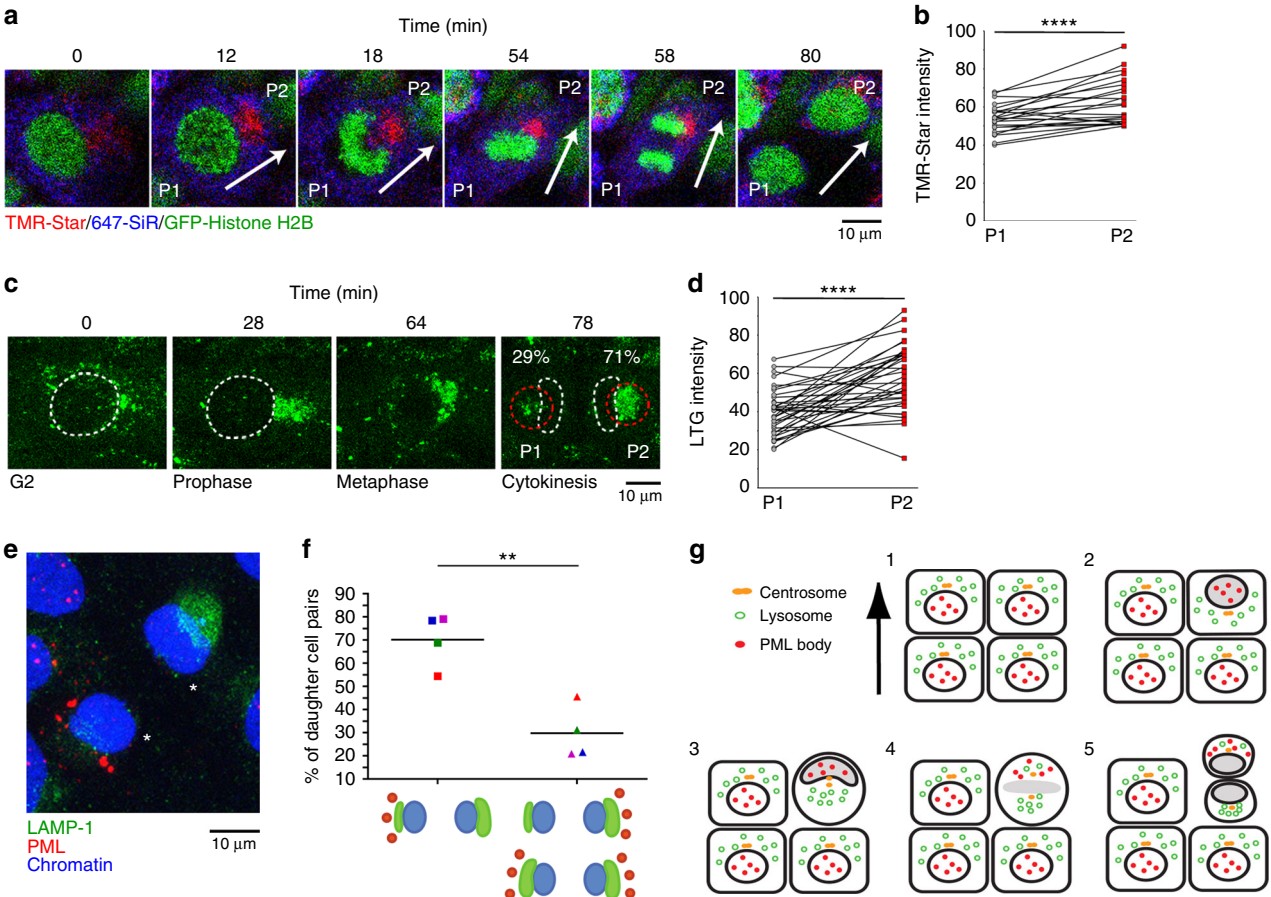

**Fig. 6** Asymmetric inheritance of lysosomes. **a** Time-lapse images of a HaCaT cell expressing Snap-tagged OMP25 and GFP-Histone H2B (green). Quiescent cells were subjected to 647-SiR (blue) and TMR-Star (red) labeling 5 h prior to and 25 h after serum stimulation, respectively. Images are extracted from Supplementary Movie 17. **b** Quantification of TMR-Star intensity in newly divided P1 and P2 daughter cells. Paired Wilcoxon's rank-sum test gives a ****$p$ value of $8.8 \times 10^{-5}$, with mean estimates of 53.92 (95% CI 49.68–58.15) and 62.77 (95% CI 58.53–67.00) for P1 and P2, respectively ($n = 23$). **c**, **d** Serum-stimulated HaCaT keratinocytes were treated with LTG immediately prior to imaging starting 25 h post serum stimulation. **c** Time-lapse images of LTG distribution during cell division. White lines indicate nuclear boundaries. Red circles indicate quantified regions. Images are extracted from Supplementary Movie 18. **d** Quantification of LTG intensity in newly divided P1 and P2 daughter cells. Paired Wilcoxon's rank-sum test gives a ****$p$ value of $1.8 \times 10^{-6}$, with mean estimates of 39.52 (95% CI 34.75–44.30) and 57.43 (95% CI 52.65–62.21) for P1 and P2, respectively ($n = 37$). **e** Distribution of LAMP-1 and PML in dividing HaCaT cells after starvation and serum stimulation for 30 h. White asterisks indicate newly divided daughter cell pair. LAMP-1 (green), PML (red), and DAPI (blue) is shown. **f** Quantification of relative LAMP-1 and PML body distribution in newly divided daughter cell pairs. A cell division was scored as asymmetric if one of the daughter cells contained the highest LAMP-1 intensity, while the other contained the majority of CyPNs. The scatter plot shows four experiments (each experiment is color-coded). Black lines indicate mean values; **$p = 0.0026$. **g** Illustration showing how the polarized cell division is aligned with cell migration, leading to segregation of PML bodies and lysosomes to the front and rear daughter cell, respectively. Black arrow indicates the direction of migration

20 h. The results revealed significantly higher velocity and Euclidean distance traveled for LTG (High) compared to LTG (Low) cells (Fig. 7f–h). Furthermore, treatment of cells with the lysotropic drugs vacuolin-1 or glycyl-L-phenylalanine 2-naphthylamide (GPN) completely reversed the cell migration phenotype seen in LTG (High) cells (Fig. 7f–h; Supplementary Movie 20). This result is in agreement with previous studies showing a role of lysosomes in cell motility[47–49].

## Discussion
In this study, we show that serum stimulation of quiescent keratinocyte monolayers generates a burst of long-range collective migration throughout the cell sheet, which is temporally and spatially coordinated with a polarized asymmetric cell division event that gives rise to daughters with differences in stemness and motility. In this system, activation of coordinated cell sheet motility does not require the presence of a wound edge as in the

traditional scratch or barrier assays. Instead, the motility barrier imposed by a restricted confluent cell sheet seems to be breached through serum-induced activation and alignment of polarized self-propulsion forces. The polarity alignment can, in part, be simulated by a Vicsek-like model, which represent one of the most commonly used models for describing flocking behavior of a wide variety of animal species and single-cell organisms[20,50–52]. However, further research is needed to identify the molecular mechanisms that regulate alignment of collectively migrating epithelial cells.

The present study also uncovers a chain of interconnected cell polarization events that begins with self-propelled polarized migration of single cells and ultimately leads to globally aligned asymmetric cell division. The global nuclear migration to the front of migrating cells is reminiscent of interkinetic nuclear migration, which has been previously described in different types of stratified epithelia[53]. It could be speculated that this behavior is crucial for proper alignment of the nucleus prior to

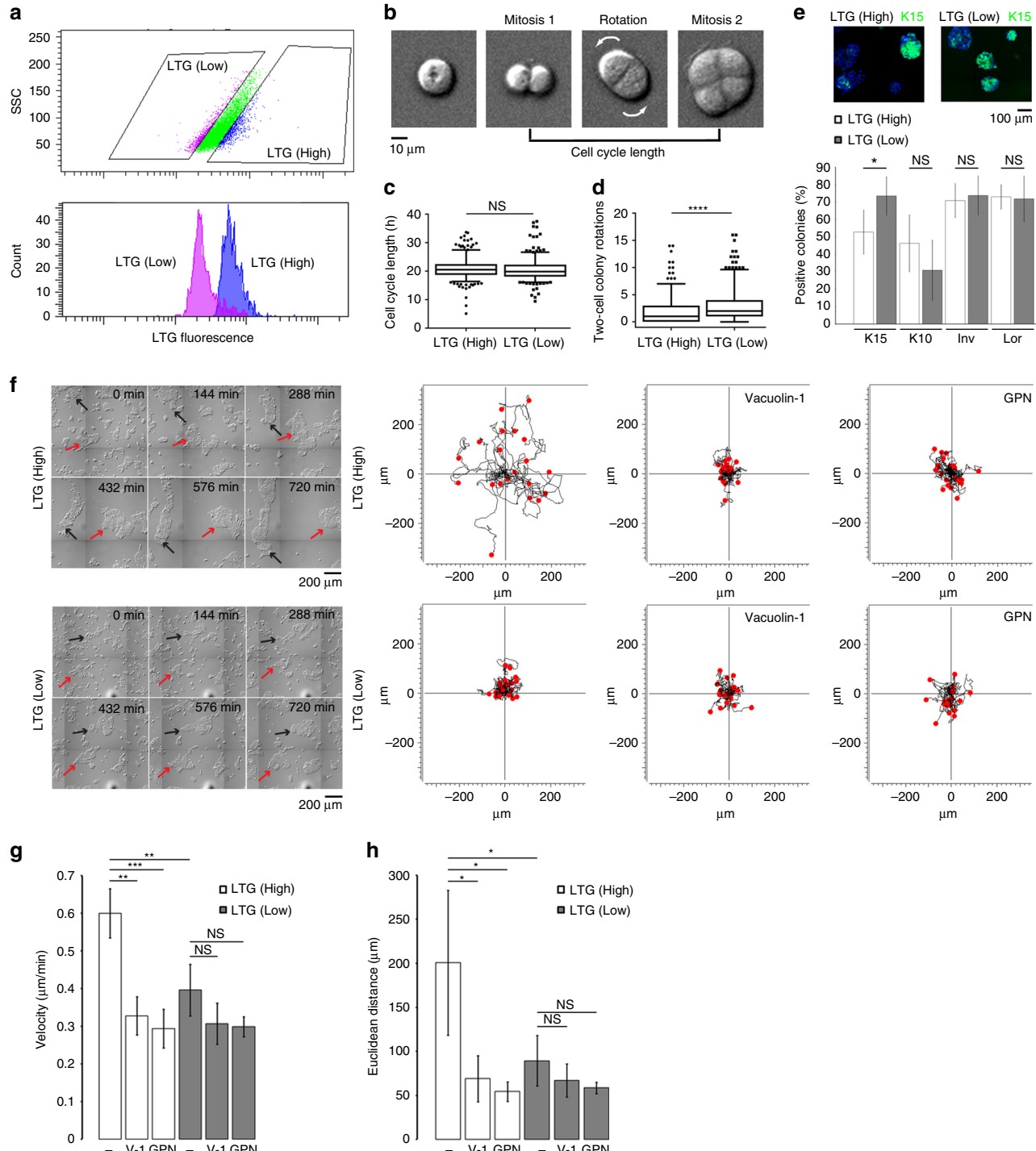

asymmetric segregation of PML bodies and lysosomes during mitosis. Interestingly, both of these organelles are strongly connected to the nucleus. While PML bodies represent integrated components of the nuclear environment, lysosomes tend to associate with the MTOC, which is closely associated with the nuclear periphery.

Given the role of PML and PML nuclear bodies in regulation of development, cell fate decision, and chromatin integrity[54–58], it is possible that asymmetric inheritance of these structures represents an additional level of control used by stem and progenitor cells to generate differential cell commitment. Inheritance of high or low levels of lysosomes, on the other hand, was found to have a

strong impact on cell motility. This result is in agreement with previous studies showing that late endosomes and lysosomes regulate cell migration through intracellular trafficking, degradation of the integrin α5β1, and regulation of calcium homeostasis[47,48,59,60]. Furthermore, EGFR was previously shown to be asymmetrically segregated in dividing primary human keratinocytes[61]. Since early endosomes previously have been shown to tether to PML bodies, plasma membrane-bound receptors, such as EGFR, may be affected by asymmetric PML body partitioning through endocytosis[62].

The present study demonstrates serum-dependent activation, long-range collective migration and globally polarized division of

**Fig. 7** Lysosome segregation regulates stemness and cell motility. **a** FACS-mediated isolation of cell populations with high and low content of LTG (upper panel). Flow Cytometry re-analysis of isolated LTG (Low) and LTG (High) fractions (lower panel). **b** Example of a single-cell undergoing mitosis and rotation at the two-cell colony stage. Images are extracted from Supplementary Movie 19. **c** Boxplot showing the time elapsed between first and second mitosis following plating of LTG (High) and LTG (Low) cells at clonal densities. Black lines are the median and whiskers indicate the 5 to 95 percentiles; $p = 0.24$; $n = 350$. **d** Boxplot showing the number of two-cell colony rotations of LTG (High) and LTG (Low) cells. Black lines indicate the median and whiskers indicate the 5 to 95 percentiles; ****$p = 9.99 \times 10^{-8}$; $n = 350$. **e** Quantification of differentiation and stem cell markers in colonies 6 days after plating. Upper panel shows representative images of LTG (Low) and LTG (High)-derived colonies stained with DAPI and anti-K15 antibodies. Graph shows quantification of colonies positive for K15, K10, involucrin (Inv) and loricrin (Lor). Bars represent mean ± SD; *$p = 0.024$; $n = 5$ experiments. **f** Analysis of cell motility. Panels to the left show representative cropped still images of isolated cell populations after plating. Red and black arrows indicate the position of two randomly selected motile colonies in the LTG (High) fraction and stationary colonies in the LTG (Low) fraction, respectively. Images are extracted from Supplementary Movie 20. Graphics to the right show motility patterns of LTG (High) and LTG (Low) cells in the presence or absence of the lysosome inhibitors vacuolin-1 or GPN after plating on collagen IV. **g** Quantification of mean velocity of fractionated cell populations in the presence or absence of vacuolin-1 (V-1) and GPN. Bars represent mean ± SD; ns, $p > 0.05$; **$p \le 0.01$; ***$p \le 0.001$; $n = 4$ experiments. **h** Quantification of mean Euclidean distance traveled by fractionated cells in the presence or absence of vacuoline-1 (V-1) and GPN. Bars represent mean ± SD; ns, $p > 0.05$; *$p \le 0.05$; $n = 4$ experiments. ns not significant

otherwise quiescent and static human keratinocytes. The migration phenotype is dependent on prior induction of dormancy, EGFR-mediated induction of self-propelled polarized forces and a putative alignment mechanism that follow the principles of the Vicsek model for flocking behavior. In addition, the present experimental system provides insight into a series of cell polarization and alignment events leading up to asymmetric cell division and selective delivery of cellular components such as PML bodies and lysosomes to front and rear daughter cells, respectively. Thus, our study provides mechanistic insight into the coordinated action of serum-induced awakening, mobilization, and division of quiescent human keratinocytes.

## Methods
**Cell lines and culturing conditions**. HaCaT (300493; CLS), NB4 (provided by Michel Lanotte)[63], and HeLa (CCL-2.2; ATCC) cells were grown in Iscove's modified Dulbecco's medium (IMDM; MedProbe) supplemented with 10% fetal bovine serum (FBS; Thermo Fisher Scientific) and 90 U/ml penicillin/streptomycin (PenStrep; Lonza). HEKn (C0015C; Life Technologies) were grown in EpiLife medium (Life Technologies) containing 60 μM CaCl₂ and supplemented with 10 ng/ml human recombinant EGF (236-EG; R&D Systems), 1 μg/ml hydrocortisone (Sigma-Aldrich), 2 μg/ml insulin, 1.1 μg/ml transferrin, and 1.34 ng/ml selenium (Life Technologies). HeLa cells were grown on retronectin-coated coverslips in order to preserve mitotic cells. In brief, coverslips were incubated with 50 μg/ml poly-D-Lysine (Millipore) for 5 min at room temperature (RT), washed with MQ water, incubated with 100 μg/ml retronectin (TAKARA) for 2 h at RT, washed with MQ water, blocked in 2% bovine serum albumin (Saveen Werner) dissolved in phosphate-buffered saline (PBS) for 30 min at RT, and finally washed and stored in PBS at 4 °C. HaCaT cells were grown either directly on glass or on collagen IV-coated surfaces. Coating of surfaces was performed by the addition of 20 μg/ml collagen IV (C7521; Sigma-Aldrich) and incubation at 4 °C overnight (ON). Starvation of HaCaT cells and subsequent re-stimulation of cell cycle progression was performed as previously described[40]. In brief, cells were cultured in serum-free medium for 2–3 days and the medium was subsequently replaced with IMDM containing 10% FBS or CNT-Prime medium containing EGF. All cell lines were tested for mycoplasma contamination.

**Epidermal cell isolation and stem cell enrichment**. After obtaining local ethical approval and informed consent from all human participants, epidermal keratinocytes were isolated from the skin from three different living donors undergoing abdominal reduction surgery (abdominoplasty). Experimental protocols for the isolation and use of epidermal keratinocytes were in compliance with all relevant ethical regulations and approved by the Regional Ethical Committee for Medicine and Health South-east Norway reference: 2013/815/REK South-east C. Pieces of dermis and epidermis layers (2 × 0.3 cm) were incubated in 1:1 dispase II (Sigma-Aldrich) and CNT-Prime medium (Cellntec) with 100 μg/ml PenStrep at 4 °C ON. The epidermis was then separated from the dermis, incubated with 0.025% trypsin/EDTA (Sigma-Aldrich) at 37 °C for 4 min, followed by neutralization using 2 mg/ml soybean trypsin inhibitor (Sigma-Aldrich). Stem cell enrichment was performed according to an established method[41]. Briefly, primary cells were seeded (25,000 cells/cm²) in serum-free CNT-Prime medium, on collagen IV-coated (2 μg/cm²; Sigma-Aldrich) plastic Thermanox coverslips (Thermo Fisher Scientific). Unattached cells were carefully removed after 20 min incubation at 37 °C and the remaining attached cells were rinsed with CNT-Prime medium. These cells were considered the stem cell enriched fraction.

**Isolation of LTG (High) and LTG (Low) cell populations**. HaCaT keratinocytes cultured in T175 flasks were subjected to serum deprivation for 2 days and subsequent serum stimulation for 40 h in order to produce a synchronous population of cells, where more than 90% of cells have passed through mitosis once. After stimulation, cells were treated with 60 nM LTG (Thermo Fischer Scientific) for 30 min and then washed twice with normal medium. Immediately after, cells were detached from flasks using trypsin, re-suspended in normal medium, and then filtered through a 100 μm Corning® cell strainer filter (CLS431752; Sigma-Aldrich). The cells were sorted into two populations, LTG (High) and LTG (Low), using a Sony SH800 cell sorter (Sony Viotechnology Inc.).

**Live cell imaging**. For live confocal microscopy of single cells, cells expressing fluorescent proteins were seeded on 35 mm glass bottom dishes from MatTek Corporation. Live imaging was carried out using a Leica TCS SP8 confocal microscope equipped with a ×40 1.30NA oil immersion lens. The microscope stage was built into an incubation chamber maintaining 37 °C and 5% CO₂. Z-stacks comprising of 8–12 images were generated at 1–4 min intervals. Projections of z-stacks and image analysis were performed using ImageJ (http://imagej.nih.gov/ij/). For imaging of large two-dimensional epithelial cell sheets, a Zeiss wide-field Axiobserver Z1 microscope controlled by Micro-Manager software (V2.0b) was used[64]. The microscope was equipped with ×10 0.3NA and ×20 0.8NA objectives, LED illumination (pE4000; CoolLED), a CMOS camera (ORCA-Flash4.0; Hamamatsu), and a high-precision stage with CO₂, humidity, and temperature control. A grid of up to 10 × 10 adjacent images was acquired. Following image acquisition, image panels were stitched together into a mosaic image, which was used for further processing. Mosaic time-lapse images were pre-processed by temporal median filtering in 3 × 4 min frame (12 min) bins, that is, each pixel value in the filtered image was the median of the pixel values from frame $n$, $n + 1$, and $n + 2$. This was done in order to reduce the spatial noise from free-floating cells. Filtering was performed using a custom ImageJ plugin, Collective Migration Buddy V1.0 (code and further documentation available at https://github.com/Oftatkofta/ImageJ-plugins). High content imaging of live cells was performed using an ImageXpress Micro Confocal microscope from Molecular Devices. Images were acquired in wide-field mode with 10–20 min time intervals, using a ×10 Plan Apo 0.5NA Nikon air objective and an environmental control gasket that maintain 37 °C and 5% CO₂. Time-lapse movies were created in the MetaXpress 6 software and further analysis performed using ImageJ and Python 3.6.4.

**Analysis of cell motility**. PIV analysis: Median filtered time lapses were processed using the PIV analyzer plugin (V1.2) in the FIJI ImageJ software package, using an 8 × 8 pixel search window and subpixel interpolation[65]. The **U** and **V** velocity component vector output images were used to calculate the magnitude of the velocity vectors, that is, speed, or root square velocity, using the formula $M = \sqrt{\mathbf{U}^2 + \mathbf{V}^2}$, where $M$ is a matrix containing the magnitudes of the velocity vectors. Finally, the velocity magnitude images, $M$, were passed through a 5 × 5 pixel spatial median filter, empty areas were masked, and the mean of each individual frame was recorded.

Time-dependent correlation length calculation: Serum-stimulated quiescent HaCaT keratinocytes grown in uncoated glass bottom dishes (35 mm; MatTek) were subjected to wide-field live imaging using a ×20 air objective (0.8NA). A tiled grid consisting of 64 (8 × 8) images were acquired at 4 min intervals between acquisitions. Stitched mosaic time-lapse images were 3-frame median temporal filtered and down sampled to reach a pixel resolution of 5–6 μm, where individual mCherry-Histone H2B nuclei are 2–3 pixels in size, and a time resolution of 12 min. Areas outside the central 35 mm diameter cover glass in the glass bottom dishes were masked, to prevent non-specific interference with the analysis. Following stitching, down sampling, and median filtering, PIV data were generated using the Python OpenPIV library (v0.20.5;[66]). For the correlation length analysis a

$32 \times 32$ pixel interrogation window with a 50% overlap was used for PIV analysis. This was experimentally determined to be the fastest reliable way to measure the velocity fields. Each PIV interrogation window covered an area of approximately $180 \times 180$ μm. The 5-$\sigma$ correlation length was defined as the largest distance, $r$, where the average angle between two velocity vectors $r$ micrometers apart was <90° with a statistical significance level of 5-$\sigma$ ($p = 3 \times 10^{-7}$). The computer algorithm used for these analyses was written in Python and is available at: https://github.com/Oftatkofta/Correlation_length_analysis_alg.

The algorithm can be described with the following steps:

1. Select each of the $N$ vectors along the top left to bottom right diagonal of the PIV output velocity vector array as $\mathbf{v}_0$.

2. For each $\mathbf{v}_0$, expand linearly, one row/column position at a time, along the cardinal directions (up/down/left/right) and calculate the angle between $\mathbf{v}_0$ and each of the vectors $\mathbf{v}_r$, at each position. Do not include masked positions, or positions outside of the array. The angles $\theta$ for each $\mathbf{v}_0$ were calculated with the formula:

$$\cos\theta = \frac{\langle \mathbf{v}_0 * \mathbf{v}_r \rangle}{\langle |\mathbf{v}_0| * |\mathbf{v}_r| \rangle}.$$

3. Record all the angles and distances between $\mathbf{v}_0$ and $\mathbf{v}_r$ for each $N$, and for each time point, $t$.

4. For each distance $r$, and time point $t$, average all the angles recorded at this distance:

$$\theta(r) = \frac{1}{N} * \sum_{i=1}^{N} \cos^{-1}\left(\frac{\langle \mathbf{v}_0 * \mathbf{v}_r \rangle}{\langle |\mathbf{v}_0| * |\mathbf{v}_r| \rangle}\right).$$

5. Compute the angular velocity correlation length at each time point. This was defined as the maximum distance where $\theta$ was <90° with a statistical significance of 5-$\sigma$:

$$C_{vv}(t) = \max_{r \to \infty}(r)\{\text{AVG}(\theta)(r) + 5 * \text{SEM}(\theta(r)) < 90°\}$$

Mean square displacement (MSD) analysis: Individual nuclei were tracked with the ImageJ plugin TrackMate V3.4.2[67]. With the following settings; Spot detection: LoG detector, Blob diameter 16 μm, No threshold, Median Filter, and subpixel localization Spots were not filtered. Spot linking: Simple LAP tracker, linking max distance 15 μm, gap closing max distance 5 μm, gap closing max frame gap: 0. Tracks were filtered as to contain at least 19 spots (=5 h of tracking data).

Five-hour MSD was calculated for each time point $t$, by averaging the square of the distance between positions of spots belonging to the same track, at $t_0$ and $t_{0+5h}$, for each track. Tracks that ended <5 h from $t_0$ were discarded:

$$\text{MSD}_{5h}(t) = \frac{1}{n}\sum_{i=0}^{n}(x_0 - x_{0+5h})^2 + (y_0 - y_{0+5h})^2.$$

The number of included tracks per time point $t$ is denoted by $n$. $x_0$ and $y_0$ denote the $x$ and $y$ positions of a spot belonging to track T at time $t_0$. $x_{0+5h}$ and $y_{0+5h}$ denote the $x$ and $y$ positions of a spot belonging to the same track T at time $t_{0+5h}$.

Motility analysis of sorted LTG (Low) and LTG (High) cells plated on collagen IV-coated glass: following isolation of LTG (High) and LTG (Low) cells by FACS, cells were seeded in glass bottom 12-well plates (MatTek) coated with collagen IV at a density of 400,000 cells per well. Following attachment of cells to the surface for 6 h, cells were placed on the environmental microscope stage. After equilibration for 1 h, DIC imaging was carried out using a ×20 0.8NA air objective. Images consisting of 16-tiled (4 × 4) fields of view were acquired using a frame interval of 8 min. After imaging, time series were stitched using the Python program StitchBuddy. Individual cells in the first frame of the movies were randomly selected by overlying a fixed grid consisting of 20 evenly spaced positions spanning the entire image. Selected cells were tracked using the manual tracking tool in ImageJ, and velocity and Euclidean distance traveled were calculated using the ImageJ plugin Chemotaxis and migration. The source code for StitchBuddy is available from the repository at: https://github.com/Oftatkofta/Correlation_length_analysis_alg.

**Quantification of cell division and migration parameters.** For cell division direction and nucleus-to-Golgi directionality, we defined each of the $n$ cell divisions/nucleus-to-Golgi directions as a unit vector $\mathbf{d}_n$ in the direction $\theta_n$ measured. Thus, each event can be described in terms of its cartesian components as $x_\eta = \cos(\theta_\eta)$ and $y_\eta = \sin(\theta_\eta)$. We can then calculate the average component along the $x$-axis $\langle x \rangle = \frac{1}{n}\sum_{i=1}^{n} x_n$ and along the $y$-axis $\langle y \rangle = \frac{1}{n}\sum_{i=1}^{n} y_n$. The average measured direction can then be described as $\langle \mathbf{d} \rangle = \langle x \rangle \hat{i} + \langle y \rangle \hat{j}$. The tendency towards common orientation axes can be defined according to the magnitude of the average cell division vector $\langle \mathbf{d} \rangle$. This magnitude $\| \langle \mathbf{d} \rangle \|$ is most simply calculated as $\| \langle \mathbf{d} \rangle \| = \sqrt{\langle x \rangle^2 + \langle y \rangle^2}$. A magnitude of 0 would show zero tendency to common directionality axes, while a magnitude of 1 would show completely consistent axial direction. The average of cell division $\langle \theta \rangle$ can be calculated either directly from the measured $\theta_n$ values or from the vector of average cell divisions $\langle \mathbf{d} \rangle$ using $\arctan\frac{\langle y \rangle}{\langle x \rangle}$. Calculation of the MPI of single cells over time was achieved by employing a running window of width = 10 successive time frames (8 min intervals between frames). For each window the MPI was calculated in the same manner as $\mathbf{d}$ for unit vectors.

**Numerical simulation and mathematical modeling.** Our numerical model is a combination of two different simulation models, namely the Self-Propelled Voronoi model for confluent cells[34], and the Vicsek model for self-propelled particles[20]. The cells are defined through a Voronoi tessellation of two-dimensional space. For every cell $i$, the position vector $\mathbf{r}_i(t)$ at time $t$ obeys the equation of motion

$$\mathbf{r}_i(t + \Delta t) = \mathbf{r}_i(t) + [\mu \mathbf{F}_i(t) + v_0 \mathbf{n}_i(t + \Delta t)]\Delta t,$$

where $\Delta t$ is the time step of propagation, $\mu$ is a mobility coefficient, $\mathbf{F}_i$ is the force on cell $i$ due to cell–cell interactions, $v_0$ is the self-propulsion speed arising from EGF stimulation, and $\mathbf{n}_i(t) = \begin{pmatrix} \cos\theta_i(t) \\ \sin\theta_i(t) \end{pmatrix}$ is the polarization vector of cell $i$ at time $t$. The multibody cell–cell forces are defined as the gradient of an energy functional, $\mathbf{F}_i(t) = -\nabla_i E$, with

$$E = \sum_{j=1}^{N} K_A(A_j - A_0)^2 + K_p(p_j - p_0)^2.$$

Here $N$ is the total number of cells, $A_j$ and $p_j$ are the surface area and perimeter of cell $j$, $A_0$ and $p_0$ are the preferred cell area and perimeter values at which the energy is minimized, and $K_A$ and $K_p$ represent the area and perimeter stiffness moduli, respectively. The first term in the energy functional arises due to volume incompressibility and the resistance of the monolayer to height fluctuations, while the second term originates from the active contractility of the cell cortex and cell–cell adhesion[34].

For the time evolution of the polarization angle $\theta_i(t)$, we use a Vicsek-like equation of motion,

$$\theta_i(t + \Delta t) = \theta_i(t) + \frac{\Delta t}{\tau_V}\left\langle \phi_j(t) - \theta_i(t) \right\rangle_{0 < |\mathbf{r}_j - \mathbf{r}_i| \leq R_V} + \eta_i(t),$$

where $\tau_V$ is a persistence time for the cell–cell alignment, $\phi_j(t)$ is the angle associated with the instantaneous velocity vector of neighboring cell $j$ at time $t$, defined by $\begin{pmatrix} \cos\phi_j(t) \\ \sin\phi_j(t) \end{pmatrix} = \frac{\mathbf{r}_j(t) - \mathbf{r}_j(t - \Delta t)}{|\mathbf{r}_j(t) - \mathbf{r}_j(t - \Delta t)|}$, the brackets $\langle \dots \rangle_{0 < |\mathbf{r}_j - \mathbf{r}_i| \leq R_V}$ denote an average over all neighboring cells $j$ that lie within a distance $R_V$, and $\eta_i(t)$ is a white noise term with zero mean and variance $2D_r$. Thus, in our model, each cell tends to align its direction of self-propulsion with the velocity direction of its neighbors over a characteristic time and in the presence of noise. The magnitude of the Vicsek radius $R_V$ is a measure for the connectivity between cells; a large $R_V$ implies that the polarization of a cell is strongly coupled to the motion of its neighbors, thus mimicking the effect of enhanced cell–cell connectivity by increased calcium concentrations.

The combined equations of motion for the cell positions $\mathbf{r}_i(t)$ and orientations $\theta_i(t)$ dictate the full cell dynamics in our model, which we solve numerically using a modification of the cellGPU code[68]. The dynamics is thus governed by a balance between the cells' tendency to minimize the energy by achieving a target cell geometry, and the cells' tendency to align their velocities through the Vicsek mechanism with stochastic Brownian noise. Note that in the limit of $\tau_V \to \infty$, neighboring cells are unable to align on any finite time scale, and our model reduces to the Self-Propelled Voronoi model[34] in which the direction of the active force $\mathbf{n}_i(t)$ undergoes only simple Brownian rotation. In the limit of $\mu = 0$ and $\tau_V = \Delta t$, we essentially recover the original Vicsek model for point-like self-propelled particles[20]. We initialize our simulation model by placing $N$ cells at random positions and with random polarization vectors in a square box with periodic boundary conditions. We equilibrate the system for 10,000 time steps without the active force ($v_0 = R_V = 0$) before starting the actual simulation of 20,000 time steps. Dimensionless parameters are obtained by measuring all length scales in units of $r_0 = \sqrt{A_0}$ and time scales in units of $\tau_0 = (\mu K_A A_0)^{-1}$. We set $N = 1000$, $K_A = K_p = 1$, $\mu = 1$, $v_0 = 0.5$, $\tau_V = 1$, $D_r = 1.5$, $\Delta t = 0.001$, and vary the value of $R_V$ from 0 to 10 to study the effect of varying calcium concentrations. For the preferred cell area and perimeter we use a ratio of $\frac{p_0}{\sqrt{A_0}} = 4.0$, close to the experimentally observed average value. The average cell speed at time is calculated by averaging $v_i(t) = \frac{\mathbf{r}_i(t + 100\Delta t) - \mathbf{r}_i(t)}{100\Delta t}$ over all cells $i$. The IOP is calculated in the same manner as in experiment, but with only one frame of view. The data presented are obtained by averaging the simulation results over 30 independent runs. While the quantitative results depend explicitly on all parameters in our model, we have verified that our qualitative findings and conclusions, in particular the observation of enhanced collective cell alignment and flocking with increasing $R_V$, also hold for different parameter choices. A detailed study on the effect of all remaining parameters in our model will be studied in a separate publication.

**Tracking and analysis of pre-mitotic nuclear migration**. Nuclear motility prior to mitosis was analyzed by live imaging of HaCaT cells stably expressing mCherry-Histone H2B following starvation and subsequent serum stimulation of cells for 25–30 h. DIC was used for detection of the plasma membrane. For each cell division three different coordinates were defined at time point zero (the last time point before mitosis entry): $c$ (the center of the nucleus), $x$ (the point on the plasma membrane closest to the nucleus). and $y$ (the point on the plasma membrane opposite $x$, defined by a straight line between $c$ and $x$). By reversing the time-lapse series, coordinates at consecutive time points, using the same criteria, were obtained at 2 min intervals between frames for a total of 2 h. Relative nuclear position was calculated as the distance $c-x$ divided by the distance $y-x$. In control experiments cells were incubated in the presence of 100 ng/ml nocodazole (Sigma-Aldrich) starting 1 h prior to imaging.

**Code availability**. The codes used for PIV-based cell motility analysis were written in Python and are available at: https://github.com/Oftatkofta/Correlation_length_analysis_algt.

Codes used for the numeric simulations are available on request from LMC Janssen at Eindhoven University of Technology, Eindhoven, Netherlands.

## Data availability

All data that support the findings of this study are available from the corresponding author on reasonable request.

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

## Acknowledgements

The work was supported by the Norwegian Cancer Society and South Eastern Norway Regional Health Authority. We would like to acknowledge Professor Kees Storm for many useful discussions. Flow cytometry was performed at Flow Cytometry Core Facility at the Oslo University Hospital, Radiumhospitalet.

## Author contributions

E.L., A.P., A.L., P.B., C.J.J., J.E., and S.O.B. performed experiments. K.A.T. provided the study with primary human epidermal cells. E.L., T.P.U., J.E., and S.O.B. designed experiments. J.E. wrote image analysis software. A.D.R. performed statistical analysis. M. V. implemented and performed the numerical simulations under supervision of L.M.C.J. E.L. and S.O.B. wrote the paper with input from all authors.

## Additional information

**Competing interests:** The authors declare no competing interests.

Additional Methods including antibodies, fluorescent reagents, and drugs; lentiviruses; immunofluoresence (IF) analysis; protein extraction and western blotting; quantification of cell division by flow cytometry; analysis of cell cycle length and two-cell colony rotation; colony size analysis; expression of stem cell and differentiation markers; and statistical analysis are described in the Supplementary Information.

