## [Peer Review File · Nature Communications]

Reviewers' comments:

Reviewer #1 (Remarks to the Author):

This manuscript from the Bøe lab develops and characterizes a new system of "wound healing" that involves serum starvation and restoration of an immortalized human keratinocyte cell line (HaCaT cells). They utilize this system to demonstrate that serum restoration induces collective migration, which is accompanied by polarized mitoses and asymmetric inheritance of PML-NBs and lysozyme. In addition, they provide evidence that the differential expression of lysozyme among daughter cells correlates with different levels of "stemness." Overall, this is a well-written, comprehensive, and multi-pronged manuscript with many strengths. There were some areas which I would have liked to see further developed (e.g. the functional role of PML asymmetry in "stemness" and use of a primary epidermal (rather than immortalized) cell line for the experiments in Fig. 5. I am also not sure I buy into the logic that serum exposure to an immortalized keratinocyte cell line is analogous to blood exposure during wounding. That said, overall the experiments are well controlled, thoughtfully analyzed, and reasonably interpreted. I particularly like the clever "rescue" experiments in Fig. 5G,H. The manuscript is generally very well written and the thorough and detailed Methods section is appreciated. I support publication pending attention to the comments below, which are largely textual rather than experimental.

Major comments:

1. In the second paragraph of the Introduction (beginning at line 57), a two-step wound healing paradigm whereby migration at the leading edge is followed by proliferation behind it is introduced. This is based on citations of two studies of corneal wound healing, largely ignoring a much larger and more relevant body of literature on epidermal wound healing. Examples of reviews that could be cited are: Safferling, J. Cell Biol. 203, 691–709 (2013); Shaw & Martin, Curr. Opin. Cell Biol. 42, 29–37 (2016). In addition, though the recent work of Park & Greco (ref. 20) was later cited, this work, based on live imaging of wound closure in murine epidermis, presents a revised model where in fact there are three zones: a migrating zone at the leading edge, a proliferating but stationary zone far from the wound front, and a mixed region in between. The oversimplified model presented here overstates the significance of the Authors' findings in this study and ignores some important recent studies. In particular, the sentence in lines 62-64 is too strong and should be removed. Moreover, the Park & Greco findings dispute the statement made in the Discussion (lines 380-381) that migration and proliferation represent "mutually exclusive states."
2. Lines 125-127: What is the rationale for choosing to inhibit EGF signaling? Perturbing Rac or nocodazole/colcemid seem like more intuitive ways to block migration. It is also not clear to me whether low-dose nocodazole treatment (used in the experiments in Fig. 2G to perturb nuclear positioning) also impacts collective migration.
3. It is stated in lines 160-161 that nocodazole "completely inhibits plasma membrane proximal positioning of chromatin during prophase." However, in Fig. 2G, the relative nuclear position in the nocodazole group at mitotic entry is still ~ 0.42 vs ~ 0.38 for untreated cells. This statement is too strong and the word "completely" should not be used. I would argue in fact that both conditions show some polarity, but nocodazole weakens this effect (although it is not tested whether this effect is significant).
4. Related to Fig. 3, I have some concerns with the way the authors choose to define "symmetry" (e.g. both daughters inherit some PML) vs. "asymmetry" (only one daughter inherits PML). For example, in lines 206-208: I disagree with the statement that inheritance of some, even if largely unequal, amounts of PML bodies between both daughter cells represents "symmetric inheritance." Symmetric implies equal (and asymmetric unequal), which is not the case here. Perhaps the phrase "dual inheritance" might be more appropriate. Moreover, this way of measuring skews heavily toward increasing the likelihood of observing asymmetry with decreasing numbers of PML bodies. For example, in Fig. S2B (starved+serum) there are a huge number (perhaps 1/3 of all cells) that have 4 or fewer total PML bodies, and almost all of these are categorized as asymmetric, while in the control group in Fig. S2A, far fewer cells have ≤ 4 PMLs. Admittedly, the

analyses in Fig. 3D are well done, but I feel that Fig S2 and 3C overstate the difference between the control and serum restored groups.

5. A model of summary figure panel depicting migrating cells with the position of the nucleus/chromatin, polarized PMLs/lysosomes and orientation of migration as mitosis progresses in starved and serum-replenished states would be very helpful.

Minor comments:

1. Line 39: "PML bodies" should be defined in the abstract
2. Line 75: I do not understand what is meant by "epithelial cells normally reseed in a quiescent state..."
3. The graphs in Fig 2A should be drawn on the same radial scale (the one on the left goes to 12 and the one on the right to 14). Also, binning in 10° rather than 5° increments may improve the data presentation.
4. In addition, for the graphs in Fig. 2I, the scales need to be normalized. In many cases the n value for the starved + serum condition is much higher, which leads to much larger bars that may exaggerate the observed effect (though I do believe there is an effect).
5. For the images portrayed in Figs. 2B-E, is this in a starved (non-migratory) or serum-restored (migratory) condition? Is the polarization of nuclear positioning
6. Lines 172-174: This is somewhat confusing wording. As I understand it, the P1 (convex) side of the chromatin faces the leading edge (front), correct? Please rephrase to make more clear. With the various arrows sometimes indicating migration direction and sometimes division vector, it was at times difficult for me to follow.
7. Lines 184-186: a reference for a review (or two) describing the function of PML and PML-NBs would be helpful here. Relevant to this study, it may also be worth mentioning the relevance of PML-NBs to heterochromatin and telomere maintenance.
8. Related to Major Comment 4, I do not find the image of the "asymmetry" in Fig. 3B to be representative of how the Authors' define asymmetry (e.g. no PML bodies in one daughter) because it would appear that the lower daughter inherits some PML bodies, albeit fewer than the upper daughter.
9. In general, the figure legends are rather wordy and could be written more concisely.

Reviewer #2 (Remarks to the Author):

This manuscript study collective migration in keratinocytes after serum depletion. The authors analyse cell migration and division during collective behaviour, showing that migration precede the high of cell proliferation, and that orientation of cell division is aligned with the direction of migration. In addition, they analyse several markers of cell polarity, finding asymmetric divisions that lead to a differential heritage of lysosomes. The characterization of cell populations with different levels of lysosomes show a differential migratory behaviour.

There are several interesting observations in this work, such as the rapid cell coordination after serum deprivation, the polarized and asymmetric division and its alignment with the direction of migration. However, the authors fail short of studying any causal relationship between all these different behaviours. For example, is the asymmetric division consequence of polarized migration? How the asymmetric division contributes to collective cell migration? How are lysosomes affecting cell migration? Etc. Although the system characterized in this manuscript has potential for future collective cell migration studies, the current work is too descriptive, lacking any mechanistic insight into collective cell migration.

Reviewer #3 (Remarks to the Author):

General comments/novelty:

The present manuscript is potentially interesting, however there are some issues that need clarification. First of all there is a novelty issue since EGFR dependent HaCaT keratinocyte migration (with wound healing) has previously been demonstrated by (Koivisto et al., 2006). In addition, asymmetrically dividing primary normal human keratinocytes (NHK) have been shown before to display asymmetric distribution of EGFR, leading to EGFR negative stem cells and EGFR positive more differentiated cells (Le Roy et al., 2010). The authors need to comment on that and it would be important to know, if the authors also observe asymmetric EGFR distribution upon cell division?

(Palibrk et al., 2014) have shown that PML bodies tether to the surface of early endosomes during mitosis of HaCaT cells. Authors show that PML bodies are inherited to the cell with less lysosomes. Does it mean the one cell inherits majority of lysosomes and the other early endosomes? Does it have an impact on EGFR dependent regulation of migration?

None of the above publications have been mentioned in the manuscript.

Point by point:

Major

1. "Thus, activation of global, large scale collective cell sheet motility requires a quiescent cell state prior to activation, the presence of serum growth factors, and a functional EGFR signaling pathway". To convincingly demonstrate the lack of collective sheet motility one would have to stimulate untreated cells with serum.
2. In case of wound healing assays the direction of movement is determined by filling of the gap. What determines the direction of collective sheet motility in case of confluent HaCaT cells stimulated with FBS, since there is no gradient of FBS/EGF present in the dish?
3. "Furthermore, treatment of cells with the lysotropic drugs vacuolin-1 or glycyl-L-phenylalanine 2-naphthylamide (GPN) completely reversed the cell migration phenotype seen in LTG(High) cells (Figures 5F-H; Movie S12), suggesting a direct role of lysosomes in cell motility." The direct role of lysosomes in cell motility is well known and established.

Minor

1. "Western blot analysis of proteins isolated after 2 days in culture revealed high LAMP-1 expression in LTG (High) cells showing that these cells maintain a high lysosome content." In my opinion this is an overstatement, more well established lysosomal markers would have to be analyzed, than just LAMP-1, in order to state that.
2. Initially only part of results from Figure 3C are discussed, followed by description of Figure 3D, E, F, together with Supplementary figures. The second part of Figure 3C is mentioned in the next paragraph only. I found it confusing for the reader.
3. There is no control experiment showing that Lapatinib and Gefitinib have actually inhibited EGFR at given concentrations in the cells (Figure 1F).
4. Figures 2A and 2I lack description of stimulation time and units used to present the data.
5. The scheme in Figure 4F suggests that there are no lysosomes in one of the daughter cells, which is not the case as shown in the quantification.
6. In Figure 1D there are no black arrows visible as mentioned in the figure legends.

Koivisto, L., G. Jiang, L. Häkkinen, B. Chan, and H. Larjava. 2006. HaCaT keratinocyte migration is dependent on epidermal growth factor receptor signaling and glycogen synthase kinase-3 α . *Experimental cell research*. 312:2791-2805.

Le Roy, H., T. Zuliani, I. Wolowczuk, N. Faivre, N. Jouy, B. Masselot, J.P. Kerkaert, P. Formstecher, and R. Polakowska. 2010. Asymmetric distribution of epidermal growth factor receptor directs the fate of normal and cancer keratinocytes in vitro. *Stem cells and development*. 19:209-220.

Palibrk, V., E. Lång, A. Lång, K.O. Schink, A.D. Rowe, and S.O. Bøe. 2014. Promyelocytic leukemia bodies tether to early endosomes during mitosis. *Cell Cycle*. 13:1749-1755.

Point-by-point response to the referees

Manuscript NCOMMS-17-17651-T

Title: Coordinated collective migration and asymmetric cell division in sheets of cultured human keratinocytes

Authors: Emma Lång *et al.*

We hereby submit our revised manuscript “Coordinated collective migration and asymmetric cell division in sheets of cultured human keratinocytes” to Nature Communications. We thank the reviewers for their constructive criticism and many excellent suggestions for improvement. All the comments have been carefully considered, and all the recommendations by the reviewers have resulted in amendments in the review as outlined below.

Reviewers' comments:

Reviewer #1 (Remarks to the Author):

This manuscript from the Bøe lab develops and characterizes a new system of “wound healing” that involves serum starvation and restoration of an immortalized human keratinocyte cell line (HaCaT cells). They utilize this system to demonstrate that serum restoration induces collective migration, which is accompanied by polarized mitoses and asymmetric inheritance of PML-NBs and lysozyme. In addition, they provide evidence that the differential expression of lysozyme among daughter cells correlates with different levels of “stemness.” Overall, this is a well-written, comprehensive, and multi-pronged manuscript with many strengths. There were some areas which I would have liked to see further developed (e.g. the functional role of PML asymmetry in “stemness” and use of a primary epidermal (rather than immortalized) cell line for the experiments in Fig. 5. I am also not sure I buy into the logic that serum exposure to an immortalized keratinocyte cell line is analogous to blood exposure during wounding. That said, overall the experiments are well controlled, thoughtfully analyzed, and reasonably interpreted. I particularly like the clever “rescue” experiments in Fig. 5G,H. The manuscript is generally very well written and the thorough and detailed Methods section is appreciated. I support publication pending attention to the comments below, which are largely textual rather than experimental.

Major comments:

1. In the second paragraph of the Introduction (beginning at line 57), a two-step wound healing paradigm whereby migration at the leading edge is followed by proliferation behind it is introduced. This is based on citations of two studies of corneal wound healing, largely ignoring a much larger and more relevant body of literature on epidermal wound healing. Examples of reviews that could be cited are: Safferling, J. Cell Biol. 203, 691–709 (2013); Shaw & Martin, Curr. Opin. Cell Biol. 42, 29–37 (2016). In addition, though the recent work of Park & Greco (ref. 20) was later cited, this work, based

on live imaging of wound closure in murine epidermis, presents a revised model where in fact there are three zones: a migrating zone at the leading edge, a proliferating but stationary zone far from the wound front, and a mixed region in between. The oversimplified model presented here overstates the significance of the Authors' findings in this study and ignores some important recent studies. In particular, the sentence in lines 62-64 is too strong and should be removed. Moreover, the Park & Greco findings dispute the statement made in the Discussion (lines 380-381) that migration and proliferation represent "mutually exclusive states."

In light of the above comments we have re-written parts of the introduction and discussion section. We have also toned down some of the statements in our original manuscript as suggested by the reviewer. We found the suggested literature to be very useful and it has been cited in the revised manuscript.

2. Lines 125-127: What is the rationale for choosing to inhibit EGF signaling? Perturbing Rac or nocodazole/colcemid seem like more intuitive ways to block migration. It is also not clear to me whether low-dose nocodazole treatment (used in the experiments in Fig. 2G to perturb nuclear positioning) also impacts collective migration.

In the present version of the paper we have clarified the rationale behind investigating EGFR. Briefly, since migration is activated by the presence of serum and since EGFR is a growth factor receptor that has previously been implicated in migration and wound healing; this protein seemed like a good candidate as a "first responder" for our serum-activated global collective migration. In the revised version, which we now present, we provide additional evidence that activation of EGFR play a key role in activation of cell motility (data presented in new Fig. 2).

We have data showing that a number of inhibitors, including the Rac1 inhibitor NSC23766, nocodazole and a variety of actin inhibitors also block motility. However, for the moment we assume that these are down-stream effects of EGFR signaling and we wish to investigate this in more detail in subsequent publications.

3. It is stated in lines 160-161 that nocodazole "completely inhibits plasma membrane proximal positioning of chromatin during prophase." However, in Fig. 2G, the relative nuclear position in the nocodazole group at mitotic entry is still ~ 0.42 vs ~ 0.38 for untreated cells. This statement is too strong and the word "completely" should not be used. I would argue in fact that both conditions show some polarity, but nocodazole weakens this effect (although it is not tested whether this effect is significant).

The reason why Nocodazole-treated cells (which are used as random controls in this experiment) deviates from 0.5 at mitosis onset is that cells are tracked backwards starting from mitosis onset and ending 2 hours prior to mitosis. Since we systematically select the side of the cell closest to the

nucleus as the “x-side” (see fig 4f) at the tracking start points, we expect deviation from 0.5 for completely random nuclear positioning. We agree, however, that our statements in the previous version of the manuscript were too strong. Consequently we have re-phrased these sentences to be less categorical.

4. Related to Fig. 3, I have some concerns with the way the authors choose to define “symmetry” (e.g. both daughters inherit some PML) vs. “asymmetry” (only one daughter inherits PML). For example, in lines 206-208: I disagree with the statement that inheritance of some, even if largely unequal, amounts of PML bodies between both daughter cells represents “symmetric inheritance.” Symmetric implies equal (and asymmetric unequal), which is not the case here. Perhaps the phrase “dual inheritance” might be more appropriate. Moreover, this way of measuring skews heavily toward increasing the likelihood of observing asymmetry with decreasing numbers of PML bodies. For example, in Fig. S2B (starved+serum) there are a huge number (perhaps 1/3 of all cells) that have 4 or fewer total PML bodies, and almost all of these are categorized as asymmetric, while in the control group in Fig. S2A, far fewer cells have ≤ 4 PMLs. Admittedly, the analyses in Fig. 3D are well done, but I feel that Fig S2 and 3C overstate the difference between the control and serum restored groups.

We agree that our use of the word symmetry to describe cell division where both cells inherit PML bodies is imprecise. We think “dual inheritance” is an excellent suggestion and have now implemented this word in the revised manuscript. We also agree that the analysis presented in Fig. S2 (Suppl. Fig. 6 in the revised version) have some inherited limitations. The problem (as the reviewer points out) is that PML aggregates are larger and fewer in primary cells compared to the HaCaT keratinocytes. This leads to a less robust statistical analysis. In the revised paper we have now edited the text to make readers aware of these potential limitations.

5. A model of summary figure panel depicting migrating cells with the position of the nucleus/chromating, polarized PMLs/lysosomes and orientation of migration as mitosis progresses in starved and serum-replenished states would be very helpful.

We fully agree. A schematic summary of the asymmetric cell division can now be viewed in Fig. 6g.

Minor comments:

1. Line 39: “PML bodies” should be defined in the abstract

We now use the full term “nuclear promyelocytic leukemia bodies” instead of just PML bodies.

2. Line 75: I do not understand what is meant by “epithelial cells normally reseed in a quiescent state...”

We can understand that our use of the word quiescence while talking about basal keratinocytes may be subjected to debate since these cells are known to be active in homeostasis. In the present publication “a quiescent state” is being used for describing the resting, slow-turnover G_0 state that characterizes most cells in tissue, including skin cells that are undergoing homeostasis. This is in contrast to the situation in most cell culture system or wounded tissue where cells are cycling more rapidly. In the revised manuscript we have taken steps to clarify this.

3. The graphs in Fig 2A should be drawn on the same radial scale (the one on the left goes to 12 and the one on the right to 14). Also, binning in 10° rather than 5° increments may improve the data presentation.

Yes, we have now recreated these graphs using 0-1 normalization and larger bins to facilitate comparison of the two panels (Fig. 4a in the revised version).

4. In addition, for the graphs in Fig. 2I, the scales need to be normalized. In many cases the n value for the starved + serum condition is much higher, which leads to much larger bars that may exaggerate the observed effect (though I do believe there is an effect).

Yes, we have now performed 0-1 normalization of the data and re-plotted the graphs (Fig. 4h and Suppl. Fig. 4b in the revised manuscript).

5. For the images portrayed in Figs. 2B-E, is this in a starved (non-migratory) or serum-restored (migratory) condition? Is the polarization of nuclear positioning

Yes, this information were lacking in our previous version. The images are from experiments where cells have been subjected to starvation and re-stimulation. This information has now been added to the figure legend.

6. Lines 172-174: This is somewhat confusing wording. As I understand it, the P1 (convex) side of the chromatin faces the leading edge (front), correct? Please rephrase to make more clear. With the various arrows sometimes indicating migration direction and sometimes division vector, it was at times difficult for me to follow.

Yes, we can see why this may be confusing for readers. Accordingly, we have made several changes to illustrations and written text to make this as clear as possible. For example, we are now consistently using the same type and color for arrows denoting migration polarity and cell division polarity.

7. Lines 184-186: a reference for a review (or two) describing the function of PML and PML-NBs

would be helpful here. Relevant to this study, it may also be worth mentioning the relevance of PML-NBs to heterochromatin and telomere maintenance.

Yes, we have now added a few references to relevant literature on PML and PML bodies.

8. Related to Major Comment 4, I do not find the image of the “asymmetry” in Fig. 3B to be representative of how the Authors’ define asymmetry (e.g. no PML bodies in one daughter) because it would appear that the lower daughter inherits some PML bodies, albeit fewer than the upper daughter.

Only the cytoplasmic bodies are representative for bodies that are inherited from mitotic cells (please see illustration in Fig. 5a and Supplemental Video 17 in current manuscript version). In Fig. 5b, PML bodies in the lower cell are detected exclusively in the nucleus and are therefore not included in the analysis. Nuclear bodies form *de novo*, beginning approximately 30 min after anaphase onset, from diffusely distributed PML or from newly synthesized PML.

9. In general, the figure legends are rather wordy and could be written more concisely.

We completely agree! Accordingly we have been through each figure legend in order to remove unnecessary wording.

Reviewer #2 (Remarks to the Author):

This manuscript study collective migration in keratinocytes after serum depletion. The authors analyse cell migration and division during collective behaviour, showing that migration precede the high of cell proliferation, and that orientation of cell division is aligned with the direction of migration. In addition, they analyse several markers of cell polarity, finding asymmetric divisions that lead to a differential heritage of lysosomes. The characterization of cell populations with different levels of lysosomes show a differential migratory behaviour.

There are several interesting observations in this work, such as the rapid cell coordination after serum deprivation, the polarized and asymmetric division and its alignment with the direction of migration. However, the authors fail short of studying any causal relationship between all these different behaviours. For example, is the asymmetric division consequence of polarized migration? How the asymmetric division contributes to collective cell migration? How are lysosomes affecting cell migration? Etc. Although the system characterized in this manuscript has potential for future collective cell migration studies, the current work is too descriptive, lacking any mechanistic insight into collective cell migration.

1) We agree that the study is descriptive. In order to provide additional mechanistic insight we have focused on understanding the underlying mechanism by which long-range global collective migration arise. Instrumental in our analysis has been our discovery that collectively migrating cell sheets can be converted to individual cell migration through manipulation of calcium concentrations and cell density. These new experiments (which are presented in a new Fig. 2) allows us to make the following major conclusions: 1) Collective migration is driven by polarized self-propelled forces generated by individual cells and 2) long-range collective migration is achieved through the ability of each cell to align their direction, speed and persistency with neighboring cells. We confirm our hypothesis by using a novel mathematical simulation model that combines the Vicsek alignment model with a Voronoi model for confluent cells (Please see New Fig. 3).

2) To improve the flow and connectivity between the migration part and the cell division part of the paper, we have primarily focused on understanding how polarity is generated at each stage from keratinocyte activation to cell division. We present new data showing that the integrins $\beta 4$ and $\alpha 6$ can be used as migration orientation markers in individual as well as collectively migrating keratinocytes (Fig 2e, Fig 4j and k and movies 9 and 10). We also present data showing that global nuclear-to-MTOC polarity is generated during the collective migration phase and that this polarity completely reverses immediately before cell division (Fig 4j and k; supplemental Fig 4d). These new data provides a previously missing link between the collective migration and cell division face. Thus, in the revised version of the paper we provide evidence for the following sequence of events: 1) Each cell obtains polarized self-propelled forces at the time of keratinocyte activation (Fig. 2 and Fig 3). As migration coordination builds up within the sheet, global nuclear polarization is established (New Fig. 4j and k). 3) During the last period before mitosis, the nucleus reverses its polarity with respect to the migration direction, while at the same time migrates to the cell front (Fig. 3, Supplementary Movie 14). 4) This leads up to a globally polarized asymmetric cell division event where components such as the nuclear PML bodies and lysosomes are becoming selectively delivered to daughter cells derived from the front and rear of the migrating cell respectively (Fig 5 and 6).

Reviewer #3 (Remarks to the Author):

General comments/novelty:

The present manuscript is potentially interesting, however there are some issues that need clarification. First of all there is a novelty issue since EGFR dependent HaCaT keratinocyte migration (with wound healing) has previously been demonstrated by (Koivisto et al., 2006). In addition, asymmetrically dividing primary normal human keratinocytes (NHK) have been shown before to display asymmetric distribution of EGFR, leading to EGFR negative stem cells and EGFR positive more differentiated cells (Le Roy et al., 2010). The authors need to comment on that and it would be important to know, if the authors also observe asymmetric EGFR distribution upon cell division?

In the revised version of our paper we have attempted to communicate more clearly that the data showing a role of EGFR is consistent with previously published work. Accordingly, we have included the Koivisto *et al.* in addition to other citations.

As suggested by the reviewer, we have investigated the distribution of EGFR in newly divided cells. However, analysis of more than 30 cell pairs in cytokinesis did not reveal evidence for asymmetric inheritance of EGFR in our experimental system. It should be noted that we detect EGFR primarily within the plasma membrane and cannot rule out the possibility that EGFR is affected by asymmetric inheritance after internalization by endocytosis.

(Palibrk *et al.*, 2014) have shown that PML bodies tether to the surface of early endosomes during mitosis of HaCaT cells. Authors show that PML bodies are inherited to the cell with less lysosomes. Does it mean the one cell inherits majority of lysosomes and the other early endosomes? Does it have an impact on EGFR dependent regulation of migration?

We agree that this paper by Palibrk has relevance to the present study. We have now investigated asymmetric partitioning of early endosomes as the reviewer suggests. We confirmed the previously published data (Palibrk *et al.*) showing that PML and early endosomes associate during mitosis. However, we have not been able to detect asymmetric segregation of early endosomes.

None of the above publications have been mentioned in the manuscript.

The paper now include citations to these papers

Point by point:

Major

1. “Thus, activation of global, large scale collective cell sheet motility requires a quiescent cell state prior to activation, the presence of serum growth factors, and a functional EGFR signaling pathway”. To convincingly demonstrate the lack of collective sheet motility one would have to stimulate untreated cells with serum.

We agree. In response to this comment we have performed additional control experiments where cells have been treated with serum-free medium at different lengths of time. The new data are presented in Fig. 1e. Serum deprivation for 2, 4 or 12 hours followed by serum stimulation does not lead to a detectable increase in migration. Some increase in collective migration can be observed following 24 hours of starvation, while the full effect is manifested only after 48 hours of starvation.

2. In case of wound healing assays the direction of movement is determined by filling of the gap. What determines the direction of collective sheet motility in case of confluent HaCaT cells stimulated with FBS, since there is no gradient of FBS/EGF present in the dish?

Yes, this is an important question that targets the essence of our work. In the present version of the paper, we present new data (displayed in Fig. 2 and 3) that addresses this question. Briefly, we have successfully managed to break down the collective migration phenotype into different levels of coordinated movements by manipulating the calcium concentration, cell density and presence or absence of serum. This has allowed us to trace cell polarity from the single-cell state to a collective state and to compare the behavior of starved and non-starved cells at the single cell-level (Fig. 2). In addition, we test our hypothesis by using mathematical modeling and numerical simulation (Fig. 3). Our data show that directional coordinated movement depends on two main principles: 1) The ability of single cells to generate a self-propelled directional force and 2) the ability of individual cells to adopt direction, speed and persistency from neighboring cells. Thus, in our model, the cells obtain directional collective movement through flocking behavior.

3. “Furthermore, treatment of cells with the lysotropic drugs vacuolin-1 or glycyl-L-phenylalanine 2-naphthylamide (GPN) completely reversed the cell migration phenotype seen in LTG(High) cells (Figures 5F-H; Movie S12), suggesting a direct role of lysosomes in cell motility.” The direct role of lysosomes in cell motility is well known and established.

The sentence has been modified to “This result is in accordance with previous studies showing a role of lysosomes in cell motility” and citations have been added.

Minor

1. “Western blot analysis of proteins isolated after 2 days in culture revealed high LAMP-1 expression in LTG (High) cells showing that these cells maintain a high lysosome content.” In my opinion this is an overstatement, more well established lysosomal markers would have to be analyzed, than just LAMP-1, in order to state that.

In the present version of the paper, we have included LAMP-2 as a second lysosome marker (see Suppl. Fig. 11b).

2. Initially only part of results from Figure 3C are discussed, followed by description of Figure 3D, E, F, together with Supplementary figures. The second part of Figure 3C is mentioned in the next paragraph only. I found it confusing for the reader.

We have re-configured this part of the paper to have a better match between text and figures.

3. There is no control experiment showing that Lapatinib and Gefitinib have actually inhibited EGFR at given concentrations in the cells (Figure 1F).

We have added a new western blot to Suppl. Fig. 1b showing activation of EGFR in the presence of serum and reduced EGFR-phosphorylation following treatment with Gefitinib and Lapatinib.

4. Figures 2A and 2I lack description of stimulation time and units used to present the data.

This information has now been added

5. The scheme in Figure 4F suggests that there are no lysosomes in one of the daughter cells, which is not the case as shown in the quantification.

We have now modified this illustration to show unequal lysosome content between the two cells instead of the presence and total absence of lysosomes.

6. In Figure 1D there are no black arrows visible as mentioned in the figure legends.

Sorry for this. This was a mistake that went un-noticed prior to submission. It has now been corrected.

Koivisto, L., G. Jiang, L. Häkkinen, B. Chan, and H. Larjava. 2006. HaCaT keratinocyte migration is dependent on epidermal growth factor receptor signaling and glycogen synthase kinase-3 α . *Experimental cell research*. 312:2791-2805.

Le Roy, H., T. Zuliani, I. Wolowczuk, N. Faivre, N. Jouy, B. Masselot, J.P. Kerkaert, P. Formstecher, and R. Polakowska. 2010. Asymmetric distribution of epidermal growth factor receptor directs the fate of normal and cancer keratinocytes in vitro. *Stem cells and development*. 19:209-220.

Palibrk, V., E. Lång, A. Lång, K.O. Schink, A.D. Rowe, and S.O. Bøe. 2014. Promyelocytic leukemia bodies tether to early endosomes during mitosis. *Cell Cycle*. 13:1749-1755.

Reviewers' comments:

Reviewer #1 (Remarks to the Author):

This heavily revised manuscript sufficiently addresses all of my previous comments, and thus is in my opinion, suitable for publication. Minor comments and corrections are listed below. The only point of disagreement I have with the manuscript in its current form is that I do not believe the new experiments added in Figure 2 add much value.

Major comments:

- I realize that Reviewer 2 requested additional "mechanistic" insights, but I personally feel the new data added in Figure 2 does not add much value to the manuscript, and in fact makes the story more dense and less readable. I am in favor of omitting it. I do find the new modeling data in Figure 3 to be interesting and useful, however.

Minor comments:

- p. 3, lines 55-56: "Migrating cell sheets involved in wound repair are mainly formed by keratinocytes derived from the basal cell layer of epidermis." While strictly true because of the use of the word "mainly," a recent paper from Fiona Watt's lab demonstrating that suprabasal cells can migrate into basal positions and dedifferentiate to repopulate wound beds is worth citing (<https://www.ncbi.nlm.nih.gov/pubmed/28504705>)
- p. 1 line 35 and p. 4 line 85: regarding use of the term "Vicsek-like alignment mechanism." Though described later in the Results section, this should be cited upon first mention (ref 30) and briefly defined as a model to explain collective cell migration of self-propelled molecules.
- p. 7, line 167: "Instantaneous Order Parameter (IOP)" is defined here in the context of Fig. 2c, but this term is also used in earlier panel 2a, so requires earlier introduction/definition.
- Line 186: "staved" should be starved
- Line 200: "re-localize" should be "re-localized"
- The IF images in Fig. 5b,g could use labels for the fluorophores
- Line 419: missing a word, such as "we" between "experiment" and "only"
- Line 495: change "dependent" to "depend"

Reviewer #2 (Remarks to the Author):

My main objections to the previous version of the manuscript were that it was too descriptive, it lacked any mechanistic insight into collective cell migration and that some intriguing observations (e.g. polarized and asymmetric divisions are aligned with the direction of migration) were left without a causal link with the process of collective cell migration.

In this new version of the manuscript the authors have added three new experiments to address these issues. Unfortunately, they do not contribute towards a better understanding of the mechanisms that drives collective cell migration, as all these "new" results have been already extensively published in other systems.

First, the authors decrease calcium levels and conclude that the collective behaviour is impaired. This is an obvious outcome of affecting cell-cell adhesion molecules, and it has been already published (just some examples Bazellières et al (2015) Nat Cell Biol 6, 6111; Tambe et al., (2011) Nat Mater 10, 469)

Second, the authors show that cell confluence affects collective cell migration. This again is an obvious and widely published observation (see examples in this review: Park et al. (2016), J Cell Sci 129, 3375). By decreasing cell density, cell-cell contact become less frequent and the cell cluster configuration, essential for collective migration, is not reached.

Third, the authors implement a Vicsek-like model to simulate collective behaviour. Again this is not new. The Vicsek model is probably the most frequent approach to model collective behaviour in general and collective cell migration in particular (see many examples in this review: Mehes and Vicsek (2014) *Interg Biol*, 6, 831). However the generation of this model does not explain the biology behind the assumptions of the model. Just one example, the model assumes an alignment between the velocities of neighbor cells; how do cells align their velocities? How is this alignment connected with the asymmetric divisions observed in this manuscript or with the polarization?

Reviewer #3 (Remarks to the Author):

I think the paper is very good now and adds interesting new findings to the field. The authors addressed also the points of the other two reviewers appropriately andn added a large body on new evidence and plausible explanations.

Response to Reviewers' comments:

Reviewer #1 (Remarks to the Author):

This heavily revised manuscript sufficiently addresses all of my previous comments, and thus is in my opinion, suitable for publication. Minor comments and corrections are listed below. The only point of disagreement I have with the manuscript in its current form is that I do not believe the new experiments added in Figure 2 add much value.

Thank you for your positive comments. It is highly appreciated! We have modified the manuscript according to your suggestions below.

Major comments:

- I realize that Reviewer 2 requested additional “mechanistic” insights, but I personally feel the new data added in Figure 2 does not add much value to the manuscript, and in fact makes the story more dense and less readable. I am in favor of omitting it. I do find the new modeling data in Figure 3 to be interesting and useful, however.

We agree that Fig. 2 impedes the readability of our paper. Accordingly, we decided to keep only the experiments that make the most essential points demonstrating 1) activation of self-propulsion forces in response to EGF, 2) emergence of cell polarization, 3) the calcium experiment that corroborates the Vicsek radius variation in the computer simulations and 4) that cell density affects cell migration persistency. We have also modified the text in this particular section to be shorter and more concise.

Minor comments:

- p. 3, lines 55-56: “Migrating cell sheets involved in wound repair are mainly formed by keratinocytes derived from the basal cell layer of epidermis.” While strictly true because of the use of the word “mainly,” a recent paper from Fiona Watt’s lab demonstrating that suprabasal cells can migrate into basal positions and dedifferentiate to repopulate wound beds is worth citing (<https://www.ncbi.nlm.nih.gov/pubmed/28504705>)

Yes, we have now included this point in the introduction section and cited the paper.

- p. 1 line 35 and p. 4 line 85: regarding use of the term “Vicsek-like alignment mechanism.” Though described later in the Results section, this should be cited upon first mention (ref 30) and briefly defined as a model to explain collective cell migration of self-propelled molecules.

We agree, this term has now been explained in the introduction of the revised paper.

- p. 7, line 167: “Instantaneous Order Parameter (IOP)” is defined here in the context of Fig. 2c, but this term is also used in earlier panel 2a, so requires earlier introduction/definition.

Yes, the term has now been explained at first mention.

- Line 186: “staved” should be starved
- Line 200: “re-localize” should be “re-localized”
- The IF images in Fig. 5b,g could use labels for the fluorophores
- Line 419: missing a word, such as “we” between “experiment” and “only”
- Line 495: change “dependent” to “depend”

Thank you. These errors have been corrected in the revised paper.

Reviewer #2 (Remarks to the Author):

My main objections to the previous version of the manuscript were that it was too descriptive, it lacked any mechanistic insight into collective cell migration and that some intriguing observations (e.g. polarized and asymmetric divisions are aligned with the direction of migration) were left without a causal link with the process of collective cell migration.

In this new version of the manuscript the authors have added three new experiments to address these issues. Unfortunately, they do not contribute towards a better understanding of the mechanisms that drives collective cell migration, as all these “new” results have been already extensively published in other systems.

Our data show that global long-range coordinated migration can be activated in static confluent epithelial cell sheets through activation of self-propulsion forces and a Vicsek-like alignment mechanism. It is widely accepted that cell confluence in the absence of open areas imposes an energy barrier on cell motility. To our knowledge, our study is the first to demonstrate that activation of self-propelled polarized motility combined with a Vicsek-like alignment mechanism is sufficient to breach this energy barrier. Moreover, we are not aware of any previous publication(s) presenting numeric simulations on such a scenario. Notably, we see no evidence for unjamming in our system, which represents an alternative mechanism for motility activation within a confluent cell sheet. Indeed, while experimental systems exhibiting evidence for unjamming report correlation lengths in the range of 100-200 μm (please see reference 18 and 19 of our revised paper), our system generates cohorts of collectively migrating cells spanning several millimeters in size.

For further reading we will refer to the same review article as cited by the reviewer in his/her comments below (Park et al. (2016), J Cell Sci 129, 3375). While the issue of mobility restriction in confluent cell sheets is thoroughly discussed in this paper, the authors do not point to any primary research paper that reports activation of collective migration in a static monolayer of restricted confluent cell sheets through activation of self-propulsion forces.

First, the authors decrease calcium levels and conclude that the collective behaviour is impaired. This is an obvious outcome of affecting cell-cell adhesion molecules, and it has been already published (just some examples Bazellières et al (2015) Nat Cell Biol 6, 6111; Tambe et al., (2011) Nat Mater 10, 469)

We agree completely that the effect of calcium on cell-cell connectivity is well known. In fact, this is the very reason why we decided to do the experiment (as we have also stated in our previous version of the manuscript). Please note that this experiment represents one of the steps towards demonstrating that motility in our system is generated through activation of self-propulsive forces. This particular experiment also allows us to compare varying degrees of connectivity between cells experimentally to the effect of Vicsek radius variation in our simulation model. We agree that the two papers mentioned by the reviewer deserve to be cited. Accordingly, we have added citations to these papers in our newly revised version of the paper.

Second, the authors show that cell confluence affects collective cell migration. This again is an obvious and widely published observation (see examples in this review: Park et al. (2016), *J Cell Sci* 129, 3375). By decreasing cell density, cell-cell contact become less frequent and the cell cluster configuration, essential for collective migration, is not reached.

- 1) Strictly speaking, we do not look at the effect of density on collective migration in these experiments. Rather we use cell tracking to analyze the effect of cell density on the ability of individual cells to migrate persistently (please see Fig. 2d of our revised paper).
- 2) The experiments are done under non-clustering (low calcium) conditions. Thus, we see no evidence of clustering in this system even when the cells are dense.
- 3) Even if clustering did occur, we disagree with the reviewer that collective migration is an obvious consequence of increased cell density. This will only occur if cells also possess a Vicsek-like alignment mechanism that allow them to coordinate migration (please see our simulation data in Fig 3c where a low Vicsek radius leads to non-coordinated motion). This is in fact also what we observe with our non-starved cells that are used as controls in our experiments (Fig. 2d, right panel). Finally, we note that other Voronoi-based simulation studies of active confluent cells that lack an explicit alignment mechanism, e.g. by assuming simple Brownian rotational diffusion for the direction of the self-propulsion forces (see e.g. Bi et al. (2016), *Phys. Rev. X* 6, 021011), do not report large-scale collective cell migration.
- 4) Of the papers cited within the review paper mentioned above (Park et al. (2016)), we find the work by Szabó *et al.* to be of particular relevance to our experiments. This work has been properly cited also in the previous version of our manuscript.

Third, the authors implement a Vicsek-like model to simulate collective behaviour. Again this is not new. The Vicsek model is probably the most frequent approach to model collective behaviour in general and collective cell migration in particular (see many examples in this review: Mehes and Vicsek (2014) *Interg Biol*, 6, 831).

We agree that the Vicsek model is widely used for modeling collective behavior. The novelty of our simulations lies in the fact that we extend the standard Vicsek model to explicitly incorporate cell confluence and multibody cell-cell interactions through the Voronoi Hamiltonian. We also note that, even though the Vicsek model is phenomenological in nature, we can directly link the Vicsek interaction radius to the cell-cell connectivity, which we control experimentally by varying the

calcium concentration. Hence, our simulation model also allows for predictions that we can test and verify directly in an experiment. We think it is important to perform these simulations in order to demonstrate for the reader that the physical properties in question actually produce a similar type of behavior as is seen in our experimental system.

However the generation of this model does not explain the biology behind the assumptions of the model. Just one example, the model assumes an alignment between the velocities of neighbor cells; how do cells align their velocities? How is this alignment connected with the asymmetric divisions observed in this manuscript or with the polarization?

These are indeed interesting questions! In fact, work is already in progress to address molecular pathways responsible for alignment in our system. Regarding the assumptions of the Vicsek model: our calcium experiments suggest that the Vicsek alignment mechanism is mediated by direct cell-cell contacts, since stronger intercell connectivities cause a cell's velocity to be more strongly enslaved to its neighboring cell velocities. Hence, the Vicsek alignment emerges rather naturally. However, more research is needed to elucidate how the alignment, and the link with asymmetric cell divisions, is encoded at the molecular level. This requires extensive work and will have to be published elsewhere.

Reviewer #3 (Remarks to the Author):

I think the paper is very good now and adds interesting new findings to the field. The authors addressed also the points of the other two reviewers appropriately and added a large body on new evidence and plausible explanations.

Thank you, we really appreciate these positive comments.

REVIEWERS' COMMENTS:

Reviewer #2 (Remarks to the Author):

There are some interesting observations in this study; unfortunately, as I have already commented in my revisions of previous versions of the manuscript no mechanistic explanation is offered for the most interesting observations.

If the authors had focused in some of the many intriguing observations to establish a mechanistic link between their treatment and the emergence of collective cell migration, I would have been more excited about this work. Here are some examples of questions whose development would have taken more interest on my part:

1. How does deprivation/addition of serum (or EGF) trigger collective cell migration? Is the effect dependent on cell proliferation? Or does the treatment increase single cell motility?
2. How does the treatment polarize integrins? And what is the role of this polarization on collective cell migration?
3. The authors conclude that their treatment induces "self-propelled polarized forces". But what does it mean? No force measurement is performed. In my understanding cell migration is ALWAYS dependent on "self-propelled forces" (actin polymerization, FA dynamic, etc). So, nothing new here.
4. What is the significance of the prophase invagination on one side of the nucleus observed in this work in relationship to collective cell migration? Is there any causal relationship or just an epiphenomenon?
5. What is the role of asymmetric PML body or lysosome inheritance in collective cell migration? This seems to be just other markers of cell polarity. Is this relevant for collective cell migration? How does the treatment induce this cell polarization if the serum is added in a "non-polarized" fashion?
6. What is the mechanism for cell alignment?

In the rebuttal letter the authors says "our study is the first to demonstrate that activation of self-propelled polarized motility combined with a Vicsek-like alignment mechanism is sufficient to breach this energy barrier". As I already mentioned before the Vicsek model is the most common framework to simulate collective migration, including cells. Furthermore, that a particular numerical model is able to simulate a cellular behavior does not mean that the model is correct, as there are many models that can reproduce the same behavior, some of which are wrong. To test the validity of a model key experiments are required; which are missing in this manuscript by lacking a more mechanistic approach.

The experiment introduced in figure 2 (calcium manipulation) does not contribute to a more mechanistic explanation of the model; and I agree with Reviewer 1 that says that "the new data added in Figure 2 does not add much value to the manuscript" .

Point by point response letter

Reviewer #2 (Remarks to the Author):

There are some interesting observations in this study; unfortunately, as I have already commented in my revisions of previous versions of the manuscript no mechanistic explanation is offered for the most interesting observations.

If the authors had focused in some of the many intriguing observations to establish a mechanistic link between their treatment and the emergence of collective cell migration, I would have been more excited about this work. Here are some examples of questions whose development would have taken more interest on my part:

1. How does deprivation/addition of serum (or EGF) trigger collective cell migration? Is the effect dependent on cell proliferation? Or does the treatment increase single cell motility?
2. How does the treatment polarize integrins? And what is the role of this polarization on collective cell migration?
3. The authors conclude that their treatment induces “self-propelled polarized forces”. But what does it mean? No force measurement is performed. In my understanding cell migration is ALWAYS dependent on “self-propelled forces” (actin polymerization, FA dynamic, etc). So, nothing new here.
4. What is the significance of the prophase invagination on one side of the nucleus observed in this work in relationship to collective cell migration? Is there any causal relationship or just an epiphenomenon?
5. What is the role of asymmetric PML body or lysosome inheritance in collective cell migration? This seems to be just other markers of cell polarity. Is this relevant for collective cell migration? How does the treatment induce this cell polarization if the serum is added in a “non-polarized” fashion?
6. What is the mechanism for cell alignment?

In the rebuttal letter the authors says “our study is the first to demonstrate that activation of self-propelled polarized motility combined with a Vicsek-like alignment mechanism is sufficient to breach this energy barrier”. As I already mentioned before the Vicsek model is the most common framework to simulate collective migration, including cells. Furthermore, that a particular numerical model is able to simulate a cellular behavior does not mean that the model is correct, as there are many models that can reproduce the same behavior, some of which are wrong. To test the validity of a model key experiments are required; which are missing in this manuscript by lacking a more mechanistic approach.

Our response:

We agree with the reviewer that any mathematical model alone is not sufficient to explain the experimental observations. Accordingly, we have toned down all claims in the manuscript that our numeric simulations unequivocally explain our experimental observations. We have also made the following note in the discussion section immediately after discussing our use of the Vicsek model:

“However, further research is needed to identify the molecular mechanisms that regulate alignment of collectively migrating epithelial cells”.

The experiment introduced in figure 2 (calcium manipulation) does not contribute to a more mechanistic explanation of the model; and I agree with Reviewer 1 that says that “the new data added in Figure 2 does not add much value to the manuscript” .